# Neural Dispersion on Graphs

**Ryien Hosseini** [1]  **Pouya Mahdi Gholami** [1]  **Filippo Simini** [2]  **Venkatram Vishwanath** [2]
**Rebecca Willett** [1 3]  **Henry Hoffmann** [1]

## Abstract

We study the problem of generating structurally diverse graphs on $N$ unlabeled vertices. Given a space of such graphs $S_N$, metric $d$, and target cardinality $k$, the objective is to construct a set $\mathcal{G} \subset S_N$ that maximizes pairwise diversity under $d$. While neural generative models may appear appealing as a solution, standard approaches require samples from a target distribution that such *dispersion* problems lack. Thus, prior work relies primarily on combinatorial or iterative search. We instead treat diversity as an explicit optimization objective, an approach we term *Neural Graph Dispersion*. An ensemble of generators is optimized under a repulsive potential, producing diverse graphs along optimization trajectories as they disperse over $(S_N, d)$, and avoiding distribution fitting and per-metric retraining entirely. Experiments show our method produces high diversity while scaling N and k an order of magnitude beyond prior work. Our source code is available at github.com/ryienh/neural-graph-dispersion.

## 1. Introduction

Many graph-related problems require developing structurally diverse sets: evaluating neural combinatorial solvers (Veličković & Blundell, 2021), assessing GNN expressivity (Wang & Zhang, 2024), or constructing graph-structured benchmarks for reasoning models (Guo et al., 2023).

However, generating such sets is challenging: classical random graph models cover limited structural families, and recently proposed iterative search procedures scale poorly with the number of vertices. At a high level, the goal is to construct a finite set of graphs that are maximally dissimilar under a chosen notion of structural distance.

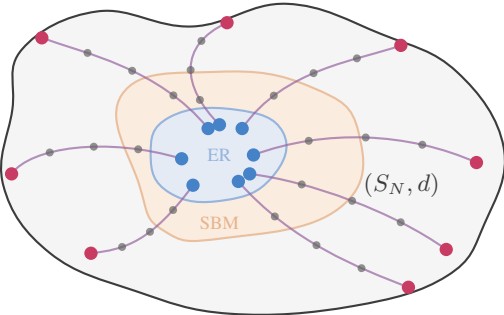

*Figure 1.* **Conceptual illustration of neural graph dispersion as particle repulsion.** We optimize a family of generators to produce diverse graphs by treating batches as repulsive particle systems in pseudometric spaces $(S_N, d)$. Particles (•) initialize near $\mathrm{ER}(N, 0.5)$ and disperse outward via gradient descent on a repulsive objective, with final positions shown in •. Graphs are sampled (•) along optimization trajectories to populate the space with structurally diverse outputs, escaping the concentration of parametric stochastic graph models such as ER and SBM.

We formalize this as the **graph dispersion problem**: given the space $S_N$ of simple graphs on $N$ unlabeled vertices, a pseudometric $d : S_N \times S_N \to \mathbb{R}_+$, and target set size $k$, our goal is to construct $\mathcal{G} = \{G_1, \ldots, G_k\} \subset S_N$ that maximizes a measure of pairwise structural diversity. For example, a standard formulation is *mean pairwise diversity*:

$$\mathcal{J}_{\text{diversity}}(\mathcal{G}; d) = \frac{1}{k(k-1)} \sum_{i \neq j} d(G_i, G_j). \qquad (1)$$

**Prior work and its limitations.** Velikonivtsev et al. (2024) recently studied the graph dispersion problem in the machine learning setting *ab initio*, proposing several approaches that fall into three categories:

- *Parametric stochastic ensembles* sample graphs from classical random graph families (e.g., Erdős–Rényi, stochastic block models) using varied parameters. While conceptually simple, this approach generally does not promote diversity: random sampling concentrates graphs in modes of the underlying distributions.

- *Combinatorial search methods* generate a large candidate pool with cardinality $M$ (where $M \gg k$) using

[1]University of Chicago [2]Argonne National Laboratory [3]NSF-Simons National Institute for Theory and Mathematics in Biology. Correspondence to: Ryien Hosseini <ryien@uchicago.edu>.

*Proceedings of the 43rd International Conference on Machine Learning*, Seoul, South Korea. PMLR 306, 2026. Copyright 2026 by the author(s).

parametric stochastic ensembles and then apply greedy or local search procedures to select a $\mathcal{G}$ via perturbation or crossover. These methods incur $\mathcal{O}(kM)$ pseudometric computations and remaining confined to the neighborhood induced by the initial seeds.

- *Generative bootstrapping* initializes a candidate pool using parametric stochastic graph models, trains a neural generator (VAEs, diffusion models) on the selected graphs, samples a large set of candidates, and greedily selects outliers to form a new training set. This process is repeated over multiple rounds to progressively expand coverage. The pipeline is doubly indirect: models are trained for in-distribution sampling but evaluated on out-of-distribution diversity, requiring repeated retraining and greedy selection at each iteration.

Ideally, one would optimize a neural model with *the explicit objective of generating diverse graphs.* However, standard neural generative models are fundamentally incompatible with this objective: Likelihood-based and adversarial approaches such as VAEs and diffusion models (Kipf & Welling, 2016; Wang et al., 2018; Xu et al., 2024) require empirical samples from a target distribution to provide supervision gradients. In the dispersion setting, no such distribution over $S_N$ exists: the goal is to construct *maximally dissimilar* graphs rather than samples from a fixed prior.

**Key insight.** We treat diversity itself as the optimization objective and introduce a neural model that induces dispersion in a graph pseudometric space. Starting from an initial collection of graphs, optimization drives their representations to separate under gradient-based repulsion, with graphs obtained by sampling states along the resulting trajectories (Figure 1). In this view, *the optimization dynamics themselves become part of the generative mechanism*, yielding candidates from distinct regions of the space. Moreover, our pseudometric induces broad structural coverage without directly optimizing individual target metrics: *a single training run yields a reusable candidate pool that can be refined under competing notions of structural similarity.* This reframing avoids distribution fitting entirely and eliminates the need for expensive, metric-dependent iterative retraining of neural models or exhaustive search procedures that sequentially explore graph candidates.

To realize this perspective, we train an ensemble of $E$ generators $\{f_{\theta_e}\}_{e=1}^{E}$, where each $f_{\theta_e} : \mathbb{R}^m \to [0,1]^{N \times N}$ maps fixed random noise vectors to edge probabilities and is optimized to promote diverse outputs. Each generator is associated with a pseudometric $d_e$, defined using a differentiable, isomorphism-invariant feature map $\phi : S_N \to \mathbb{R}^q$ based on adjacency spectral moment statistics (Section 3.3.) As points repel along independent directions during optimization of $\theta := \{\theta_e\}_{e=1}^{E}$, the union of their trajectories yields

graphs with broad structural coverage without explicitly optimizing any single downstream graph distance. We term this approach **neural graph dispersion**.

Our contributions are as follows:

- **Direct neural dispersion.** We introduce *neural graph dispersion*, the first neural framework that generates structurally diverse graphs via direct optimization, treating the training dynamics themselves as the generative process and avoiding distribution-fitting altogether.

- **Differentiable surrogate pseudometrics.** We propose a spectral-moment feature basis for graphs and show that random projections of this basis induce a family of stably differentiable pseudometrics that serve as effective surrogates for structural dissimilarity.

- **Flexible refinement under bespoke distances.** We show that neural generation produces an initial, broadly diverse set of graphs that can be efficiently refined under user-specified, non-differentiable graph distances via a lightweight selection stage, enabling adaptation to downstream notions of structural similarity.

- **Scalable graph dispersion.** We demonstrate that neural graph dispersion scales favorably in both node size $N$ and target set cardinality $k$ and thus enables graph-level dispersion at an order of magnitude larger $N$ and $k$ than previously accessible via prior methods.

- **Downstream behavioral coverage.** We demonstrate that NGD's structural coverage is functionally meaningful: on both a learned graph regressor (GNN regression) and a classical heuristic (greedy coloring), NGD-generated graphs expose performance regimes that standard parametric ensembles fail to reach.

## 2. Background

### 2.1. Preliminaries

We consider simple, undirected graphs on $N$ unlabeled vertices. Let $S_N$ be the set of such graphs, where $|S_N| = 2^{\binom{N}{2}} = \mathcal{O}(2^{N^2})$. Each graph $G \in S_N$ is represented by its adjacency matrix $A \in \{0,1\}^{N \times N}$, where $A_{ij} = 1$ indicates an edge between vertices $i$ and $j$, and $A_{ij} = 0$ otherwise. We assume $A$ is symmetric with zero diagonal. Given two graphs $G, G' \in S_N$, we write $G \cong G'$ if they are isomorphic, i.e., if there exists a vertex permutation $\pi$ such that $A_{ij} = A'_{\pi(i)\pi(j)}$. For optimization purposes, we also consider a *relaxed graph representation* in which the adjacency matrix takes values in $[0,1]^{N \times N}$. Such a matrix $\widetilde{A}$ is interpreted as a collection of edge probabilities and induces a relaxed graph $\widetilde{G}$. Discrete graphs in $S_N$ are obtained from $\widetilde{G}$ via standard discretization, e.g., edge Bernoulli sampling.

## 2.2. Graph Distance Metrics

The construction of efficient and expressive metrics over $S_N$ is a classical problem that is subject to significant study[1]. Formally, one seeks a pseudometric[2] $d : S_N \times S_N \to \mathbb{R}_{\geq 0}$ that is invariant under graph isomorphism, i.e., given graphs $G, G'$, $G \cong G' \Rightarrow d(G, G') = 0$. Each metric $d \in \mathcal{D}$ induces a corresponding metric space $\mathcal{M}_d := (S_N, d)$ over $S_N$. Throughout the paper, we view graph dispersion as the problem of generating subsets of $S_N$ that are well-separated in $\mathcal{M}_d$ under an appropriate notion of pairwise diversity.

**Evaluation metrics.** In this work, we focus on descriptor-based graph metrics for evaluating structural diversity (Vishwanathan et al., 2010), which define a Euclidean mapping $\phi(G) : S_N \to \mathbb{R}^q$ and define $d_{\text{eval}}(G, G') = \|\phi(G) - \phi(G')\|$. We defer broader discussion of graph distance constructions to Appendix A. Following prior work (Velikonivtsev et al., 2024), we report results using the graphlet correlation distance $d_{\text{GC}}$ (Yaveroğlu et al., 2014), the NetLSD metrics based on the heat and wave kernels, $d_{\text{heat}}$ and $d_{\text{wave}}$ (Tsitsulin et al., 2018), and portrait divergence $d_{\text{PD}}$ (Bagrow & Bollt, 2019). These metrics capture complementary structural properties, including local motif statistics, spectral diffusion behavior, and degree–distance distributions. We adopt standard definitions for all metrics and provide formal specifications in Appendix A. Our choice of evaluation metrics mirrors those used in prior work to enable direct and unbiased comparison across methods.

**Differentiable graph metrics.** While the above metrics are effective evaluation tools, they are often non-differentiable, computationally expensive, or tailored to specific structural statistics, limiting their use as objectives for gradient-based graph generation. Instead Section 3, introduces a differentiable graph metric constructed from spectral moment features. This metric is induced by a smooth, isomorphism-invariant feature map $\phi : S_N \to \mathbb{R}^q$ based on normalized adjacency spectral statistics. Section 4 shows that optimizing repulsive objectives defined in terms of this metric produces graph sets that exhibit high diversity under a broad range of established graph distances. Formal definitions and theoretical properties are deferred to Section 3.

## 2.3. Diversity Objectives

Given a metric space $\mathcal{M}_d = (S_N, d)$ and a target cardinality $k$, a diversity objective is a function $\mathcal{J}$ that assigns a scalar value to any $k$-element set $\mathcal{G} = \{G_1, \ldots, G_k\} \subset S_N$, typically as a function of the induced pairwise dis-

---

[1]We refer interested readers to the excellent survey by Tantardini et al. (2019).

[2]We adopt a standard abuse of notation and henceforth refer to $d$ as a *metric* on $S_N$, though in general $d(G, G') = 0 \not\Rightarrow G \cong G'$.

tances $\{d(G_i, G_j)\}_{i \neq j}$. Designing diversity objectives that are both theoretically well-founded and computationally tractable is an open problem (Mironov & Prokhorenkova, 2025). A plethora of diversity objectives have been proposed, including average (Alhijawi et al., 2022; Ruiz et al., 2023) and extremal pairwise distances (Xie et al., 2023), energy-based objectives (Velikonivtsev et al., 2024), determinantal point processes (Kulesza & Taskar, 2012), and coverage-based formulations (Xie et al., 2023).

In this work, we focus on the graph dispersion problem under a fixed choice of metric space $\mathcal{M}_d$ and diversity objective $\mathcal{J}$ and do not attempt to resolve the broader question of identifying an optimal diversity objective. For *evaluation*, we adopt the *mean pairwise diversity* objective $\mathcal{J}_{\text{diversity}}$ introduced in Equation 1 as it is widely used in practice and enables direct comparison with prior work.

For *optimization*, we employ an energy-based objective (Velikonivtsev et al., 2024) inspired by Coulomb repulsion,

$$\mathcal{J}_{\text{energy}}(\mathcal{G}; d) = -\frac{1}{k(k-1)} \sum_{i \neq j} \frac{1}{\left(d(G_i, G_j) + \varepsilon\right)^{\gamma}}, \quad (2)$$

where $\varepsilon > 0$ ensures numerical stability and $\gamma > 0$ controls the strength of short-range repulsion.

$\mathcal{J}_{\text{energy}}$ is ill-suited for *evaluation*, as it is unbounded and governed by the minimum pairwise distance: a single duplicate graph suffices to drive the objective to $-\infty$, masking the contribution of all other distances. However, such an objective is desirable for optimization, as it induces strong short-range repulsion that prevents collapse and encourages dispersion under convergent training dynamics.

## 2.4. Prior Work

The problem of selecting diverse subsets from a larger domain, or *dispersion*, has a long history in optimization and combinatorics. Classical formulations include the $p$-dispersion and max–sum dispersion problems, which seek to select $k$ elements from a metric space so as to maximize extremal or average pairwise distances (Erkut, 1990; Ravi et al., 1994). These problems have been studied extensively, with established hardness results and range of algorithmic trade-offs (Masin & Bukchin, 2008; Gao et al., 2022).

In contrast to classical dispersion formulations, the *graph-level dispersion* problem considered here has received comparatively little attention, in part due to the size and combinatorial complexity of $S_N$. Recently, Velikonivtsev et al. (2024) presented the first systematic study of graph dispersion in the machine learning setting, formalizing the problem and evaluating several algorithmic methods. Their results show that graph-level dispersion is feasible at small scales, but with strategies that become increasingly expensive as $|S_N|$ grows. We briefly summarize them below.

**Parametric stochastic graph ensembles.** A common baseline for graph dispersion samples graphs from parametric stochastic model families with varied parameters. In our experiments, we consider two such ensembles: (i) *ER-mix*, which samples Erdős–Rényi graphs across a fixed set of edge probabilities (Erdős & Rényi, 1959), and (ii) *Generator-mix*, a mixture of standard generators, including ER-mix, preferential attachment (Barabási & Albert, 1999), Holme–Kim (Holme & Kim, 2002), a power-law degree model (Aiello et al., 2001), and stochastic block models (Holland et al., 1983). Because independent draws concentrate near the modes of these distributions, such ensembles score poorly under $\mathcal{J}_{\text{energy}}$ and $\mathcal{J}_{\text{diversity}}$, which reward extremal, mutually repulsive configurations.

**Combinatorial search methods.** Velikonivtsev et al. (2024) present a class of approaches constructs diverse graph sets via explicit combinatorial search over candidate graphs. These methods maintain or iteratively explore a graph candidate pool $\mathcal{C} \subset S_N$ of size $M := |\mathcal{C}|$ generated from parametric stochastic graph models. Given the target cardinality $k$, diversity is achieved by selecting a subset $\mathcal{G} \subset \mathcal{C}$ that maximizes a chosen dispersion objective using greedy selection, genetic algorithms, or local search procedures. We outline these algorithms in detail in Appendix A. Such methods can be effective at small scales, where nontrivial regions of the metric space can be explicitly explored. However, maintaining high dispersion implicitly requires the candidate pool $\mathcal{C}$ to grow so as to cover a non-negligible portion of $S_N$. Because selection entails $\mathcal{O}(kM)$ distance evaluations, this growth causes the computational cost to escalate rapidly as $|S_N|$ increases exponentially with respect to $N^2$ (Subsection 2.1). Consequently, combinatorial search methods are inherently constrained by their reliance on increasingly exhaustive enumeration of graph space.

**Generative bootstrapping pipelines.** Velikonivtsev et al. (2024) also propose dispersion via neural generative models within an iterative bootstrapping loop. Because dispersion does not correspond to sampling from a fixed target distribution over $S_N$, standard likelihood-based or adversarial training objectives are not directly applicable. Instead, they introduce *iterative graph generative modeling* (IGGM), which alternates between fitting a generative model (e.g., Vignac et al. (2023)) to a diverse set and using the trained model to propose new candidate graphs. Concretely, a generator is trained on a candidate set $\mathcal{C}_t \subset S_N$, a graph pool $\mathcal{G}_t$ is sampled from the model, and a diverse subset is selected (via greedy selection) to form the next training set $\mathcal{C}_{t+1}$. While this procedure enables the use of expressive neural generators, it is inherently indirect: The generator is trained to fit $\mathcal{C}_t$ yet judged by the outliers it produces. Additionally, each iteration's retraining and selection over large pools limits scalability as $N$ or $k$ grows.

## 3. Neural Dispersion on Graphs

We now introduce **neural graph dispersion**, a novel neural framework for directly optimizing graph-level diversity. We formulate graph generation as a continuous optimization problem and optimize a repulsive objective (Equation 2) that encourages graphs to separate in a graph metric space $\mathcal{M}_d$. We then collect graphs along the resulting optimization trajectories, yielding a diverse candidate set. This formulation enables direct gradient-based optimization of diversity without requiring samples from a target distribution (Figure 1). Our approach consists of the following components:

- **Neural generator.** An ensemble of generators $\mathcal{F} = \{f_{\theta_e}\}_{e=1}^{E}$, where each $f_{\theta_e} : \mathbb{R}^m \to [0,1]^{N \times N}$ maps fixed noise vectors $\{z^{(i)}\}_{i=1}^{b}$ to edge probabilities and thus induces graphs $\{G_e^{(i)}\}_{i=1}^{b}$ via standard sampling.

- **Spectral-moment feature map.** A differentiable, isomorphism-invariant feature map $\phi : S_N \to \mathbb{R}^q$ based on normalized spectral moment statistics, which serves as a general structural representation of graphs.

- **Directional graph metrics.** A set of random projection operators $\{U_e\}_{e=1}^{E}$, with $U_e \in \mathbb{R}^{q_e \times q}$, that induce directional graph metrics $d_e(\widetilde{G}, \widetilde{G}') := \|U_e(\phi(\widetilde{G}) - \phi(\widetilde{G}'))\|$. Each generator $f_{\theta_e}$ is optimized to promote dispersion under the corresponding $d_e$, yielding complementary coverage across projection subspaces.

- **Bounded candidate selection.** To obtain a final set of $k$ discrete graphs for evaluation, we construct a candidate pool $\mathcal{C}$ of fixed size, independent of $N$ and $k$, from the learned generators and apply a greedy selection procedure under a chosen evaluation metric.

The remainder of this section details and analyzes the individual choices for $\mathcal{F}$, $\phi$, $d$, as well as details the optimization and bounded candidate selection. Note that for brevity, proofs for all propositions are deferred to Appendix C. Appendix D discusses alternative designs that performed suboptimally in our experiments.

### 3.1. Trajectory-Based Graph Generation

We formalize neural graph dispersion as a direct optimization approach to the graph dispersion problem. Given metric space $\mathcal{M}_d = (S_N, d)$ and target cardinality $k$, our goal is to generate a set $\mathcal{G} = \{G_1, \ldots, G_k\} \subset S_N$ whose elements are maximally separated under $d$, as measured by pairwise dispersion (Section 2.3). We achieve this by optimizing a fixed set of graphs directly in $\mathcal{M}_d$.

**Optimization objective.** To this end, we optimize the Coulomb-style energy objective $\mathcal{J}_{\text{energy}}$ defined in Equation 2. During training, this objective is evaluated on minibatches

of relaxed graphs. Specifically, given a minibatch $\widetilde{\mathcal{G}} = \{\widetilde{G}_1, \ldots, \widetilde{G}_b\}$, we minimize $\mathcal{L}(\widetilde{\mathcal{G}}; d) := -\mathcal{J}_{\text{energy}}(\widetilde{\mathcal{G}}; d)$. Throughout, $\mathcal{L}$ is optimized with respect to a differentiable graph generator (Section 3.2) and metric (Section 3.3). As discussed in Section 2.3, while Coulomb-style repulsion is ill-suited for *evaluation* due to its unboundedness, it is well suited for *optimization*: near-duplicate graphs induce strong gradients that rapidly separate degenerate configurations during training. We formalize this observation below.

**Proposition 3.1** (Stability and repulsion of Coulomb objectives)**.** *Assume $d(\cdot, \cdot)$ is differentiable on the relaxed graph domain and that its gradient is bounded on compact sets. If there exists $\delta > 0$ such that $d(G_i, G_j) \geq \delta$ for all $i \neq j$, then the Coulomb loss $\mathcal{L}(\mathcal{G}; d)$ and its gradients with respect to the generated graphs are finite and uniformly bounded. Conversely, as $d(G_i, G_j) \to 0$ for any pair $(i, j)$, the repulsive gradient $|\nabla_{G_i} \ell_{ij}|$ grows as $(d(G_i, G_j) + \varepsilon)^{-(\gamma+1)}$ grows, approaching $\infty$ when $\varepsilon = 0$ provided $|\nabla_{G_i} d(G_i, G_j)|$ remains bounded away from zero.*

**Generation as an Optimization Process.** We interpret the elements of $\widetilde{\mathcal{G}}$ as points embedded in the metric space $\mathcal{M}_d$. Under gradient-based optimization of $\mathcal{J}_{\text{energy}}$, these points follow repulsive trajectories that progressively explore distinct regions of graph space. Crucially, *graphs are generated during training by sampling along these optimization trajectories*. As particles disperse through $\mathcal{M}_d$, states are collected to form a candidate pool, making the optimization dynamics the generative mechanism. This perspective is illustrated in Figure 1 and distinguishes our approach from standard generative modeling, which requires empirical samples and is thus incompatible with dispersion tasks.

Algorithm 1 summarizes this procedure. Each expert independently optimizes a minibatch of graphs under its associated surrogate metric, inducing repulsive trajectories in $\mathcal{M}_d$. We sample uniformly from the relaxed graph representations produced by all experts to populate set $\psi$, which is then discretized into a candidate pool of graphs $\mathcal{C}$, $M := |\mathcal{C}|$. Our experiments show that a fixed pool budget $M$ suffices across graph sizes $N$ and target cardinalities $k$. Importantly, the dominant computational cost lies in generating $\mathcal{C}$, which can be amortized across downstream evaluation metrics: once constructed, the same candidate pool can be reused with different choices of $d_{\text{eval}}$ without retraining the generators. This distinguishes our approach from prior neural methods that require retraining for each new $d_{\text{eval}}$.

### 3.2. Neural Graph Generator Architecture

We parameterize graph generation using an ensemble of neural generators $\mathcal{F} = \{f_{\theta_e}\}_{e=1}^{E}$, where each $f_{\theta_e} : \mathbb{R}^m \to [0, 1]^{N \times N}$ is a multi-layer perceptron (MLP) that maps a noise vector $z \in \mathbb{R}^m$ to a *relaxed adjacency matrix*. For each $e \in [E]$, we draw a batch of fixed $b$ noise vectors

---

**Algorithm 1** Neural Graph Dispersion

1: **Inputs:** $N, k, E$, training steps $T$, collection period $r$, pool budget $M$, surrogate metrics $\{d_e\}_{e=1}^{E}$, evaluation metric $d_{\text{eval}}$
2: Initialize expert neural networks $\{f_{\theta_e}\}_{e=1}^{E}$ and fixed noise vectors $\{z^{(i)}\}_{i=1}^{b} \sim p(z)$
3: $\psi \leftarrow \emptyset$
4: **for** $t = 1, \ldots, T$ **do**
5:    **for** $e = 1, \ldots, E$ **(experts can run in parallel) do**
6:       Generate a minibatch of relaxed graphs $\{\widetilde{G}_e^{(i)}(t)\}_{i=1}^{b}$ from $f_{\theta_e}$ using $\{z^{(i)}\}_{i=1}^{b}$
7:       Update $\theta_e$ by minimizing $\mathcal{L}\left(\{G_e^{(i)}(t)\}_{i=1}^{b}; d_e\right)$
8:    **end for**
9:    **if** $t \equiv 0 \pmod{r}$ **then**
10:       Collect $\widetilde{G}$ from all experts and add to $\psi$
11:    **end if**
12: **end for**
13: **Candidate selection:** sample $M$ discrete graphs from $\psi$ and greedily select $\mathcal{G}$ with $|\mathcal{G}| = k$ under $d_{\text{eval}}$
14: **Return** $\mathcal{G}$

---

$\{z^{(i)}\}_{i=1}^{b} \sim p(z)$ (e.g., $p(z) = \mathcal{N}(0, I)$), and define

$$\widetilde{A}_e^{(i)} = f_{\theta_e}\big(z^{(i)}\big) \in [0, 1]^{N \times N}.$$

Each $\widetilde{A}_e^{(i)}$ is interpreted as a weighted adjacency matrix and induces a relaxed graph $\widetilde{G}_e^{(i)}$ used for differentiable optimization.[3] This construction implies that the dispersion does not begin from an arbitrary region of graph space, but from a restricted latent-variable random graph family, from which optimization trajectories subsequently escape (Figure 1). We formalize this intuition below.

**Proposition 3.2** (Initialization induces a restricted random graph family)**.** *At random initialization, graphs generated by $f_\theta$ and sampled via independent Bernoulli draws from the predicted edge probabilities form a latent inhomogeneous Erdős–Rényi family: conditional on the latent noise $z$, edges are independent with probabilities given by $f_\theta(z)$.*

### 3.3. Spectral-Moment Feature Map

We now define the differentiable feature map $\phi$ that serves as the structural representation used to define our graph distances. During optimization, we apply $\phi$ to relaxed graphs $\widetilde{G}$ represented by soft adjacency matrices $\widetilde{A} \in [0, 1]^{N \times N}$. For $r \geq 1$, the spectral moment $\text{tr}(\widetilde{A}^r)$ is a differentiable relaxation of the closed-walk count of length $r$ and captures structural statistics at increasing ranges. To improve numerical conditioning and reduce sensitivity to density, we use

---

[3]As we consider simple, undirected graphs (Section 2.1), in practice the generator outputs only the upper-triangular entries of $\widetilde{A}_e^{(i)}$ which are then symmetrized to form the adjacency matrix.

normalized moment features of the form

$$\mu_r(\widetilde{A}) \;=\; \frac{\mathrm{tr}(\widetilde{A}^r)}{\left(\mathrm{tr}(\widetilde{A}^2) + \delta\right)^{r/2}},$$

where $\delta > 0$ is a stability constant.[4] We then define $\phi(\widetilde{A}) \in \mathbb{R}^q$ via concatenation of fixed moment-based features:

$$\phi(\widetilde{A}) \;=\; \left[\mu_{r_1}(\widetilde{A}), \dots, \mu_{r_q}(\widetilde{A})\right] \in \mathbb{R}^q.$$

The resulting embedding $\phi$ induces a metric $d(G, G') := \|\phi(G) - \phi(G')\|$ on $S_N$. By construction, this distance is invariant to vertex relabeling and, in the limit of sufficiently rich spectral statistics, is complete up to cospectral graphs. We formalize these properties below.

**Proposition 3.3** (Isomorphism invariance of spectral features). *If $G \cong G'$ then $\phi(G) = \phi(G')$.*

**Proposition 3.4** (Completeness of spectral moments up to cospectrality). *Let $A$ be the adjacency matrix of a graph $G$. Then the moment sequence $(\mathrm{tr}(A^1), \dots, \mathrm{tr}(A^N))$ uniquely determines the spectrum of $A$ (with multiplicity), and conversely. Thus, any graph feature map from which these moments can be recovered is complete up to cospectrality.*

**Directional Metrics and Ensemble Dispersion** While $\phi(G) \in \mathbb{R}^q$ provides a general structural representation, directly optimizing dispersion under the full feature geometry can be unstable and may overemphasize particular coordinates. We therefore induce a *family* of graph distances by projecting features onto multiple random subspaces.

For each $e \in [E]$, let $U_e \in \mathbb{R}^{q_e \times q}$ be a fixed random projection operator. We define the induced metric

$$d_e(G, G') \;:=\; \left\| U_e\big(\phi(G) - \phi(G')\big) \right\|_1.$$

Each generator $f_{\theta_e}$ is optimized independently using the energy objective $\mathcal{L}_{\mathrm{energy}}(\cdot; d_e)$ on its batch $\mathcal{G}_e = \{G_e^{(i)}\}_{i=1}^b$. Intuitively, each $d_e$ emphasizes distinct spectral directions, and the union of outputs across $e$ yields structural coverage that is not tied to any single downstream evaluation metric.

**Proposition 3.5** (Separation under random projections). *Let $x, y \in \mathbb{R}^q$ with $x \neq y$, and let $U \in \mathbb{R}^{q_e \times q}$ be a random matrix drawn from a continuous distribution. Then $Ux \neq Uy$ almost surely.*

Proposition 3.5 implies that, for any pair of graphs $G, G'$ with $\phi(G) \neq \phi(G')$, a randomly drawn $U_e$ yields $d_e(G, G') > 0$ with probability one.

### 3.4. Candidate Selection

The optimization procedure in Section 3.1 produces a collection of relaxed adjacency matrices $\{\widetilde{A}_e^{(i)}\}_{i=1}^M \in [0,1]^{N \times N}$.

---

[4]Minor stabilization and implementation details are deferred to Appendix B.

To evaluate diversity under standard graph metrics, these relaxed representations must be discretized.

**Temperature sampling.** We construct a finite candidate pool $\mathcal{C} \subset S_N$ by discretizing each $\widetilde{A}_e^{(i)} \in \psi$ via Bernoulli sampling at a small, fixed set of temperatures. Specifically, letting $\sigma(\cdot)$ denote the logistic sigmoid, for a temperature set $\mathcal{T} = \{\tau_1, \dots, \tau_L\}$ with constant $L$, we sample $A_e^{(i,\ell)} \sim \mathrm{Bernoulli}\big(\sigma(\widetilde{A}_e^{(i)}/\tau_\ell)\big)$ and add the resulting graphs to $\mathcal{C}$. Varying the temperature induces controlled stochasticity, yielding multiple discrete realizations of each $\widetilde{A}_e^{(i)}$ while keeping $|\mathcal{C}|$ bounded and independent of $N$.

**Candidate selection.** Since $|\mathcal{C}| > k$ in general, we extract a final $k$-sized set by applying a selection procedure with respect to a chosen evaluation distance $d_{\mathrm{eval}}$. This stage applies a temperature-aware greedy selection algorithm in order to form $\mathcal{G} \subset \mathcal{C}$ of size $k$. The precise algorithm is described in Appendix B. This selection stage serves only to align the output with the evaluation distance and does not affect training, allowing neural graph dispersion to produce an initial diverse collection that can be refined under different, potentially competing notions of graph distance.

### 3.5. Computational Complexity and Scaling

We summarize the complexity of neural graph dispersion with respect to graph size $N$ and target set cardinality $k$.

**Proposition 3.6** (Computational Complexity with respect to $N$ and $k$). *For fixed model architecture and optimization hyperparameters, the cost of a single gradient update in neural graph dispersion scales as $O(N^3)$, with no dependence on the target set cardinality $k$.*

While the optimization cost scales polynomially with $N$, neural graph dispersion avoids dependence on graph space size $|S_N| = 2^{\binom{N}{2}}$. The neural optimization phase produces a candidate pool $\mathcal{C}$ whose size is bounded independently of $N$ and $k$. A final greedy selection step (Section 3.4) then incurs a cost of $O(kM)$, comparable to prior work, but critically without requiring $M$ to grow as the graph size increases. In contrast, combinatorial search methods rely on candidate sets whose size must expand to maintain coverage as $N$ grows (Velikonivtsev et al., 2024), leading to rapidly increasing computational cost (Appendix A).

Additionally, our proposed method consists entirely of dense, vectorized operations amenable to GPU execution; as in other domains where differentiable, accelerator-friendly formulations enabled major scaling gains, this design supports substantially larger values of $N$ and $k$ than enumeration or search-based methods (Section 4, Appendix A). As a result, dispersion is feasible in regimes where serial baselines become computationally impractical.

*Table 1.* Small-scale graph dispersion results under standard evaluation metrics. Higher values indicate greater structural diversity. Results are averaged over 5 independent runs; error bars (Appendix D) are small relative to inter-method differences. Best results per metric and graph size are shown in bold. Ablations remove either the ensemble of generators (*no ensemble*) or directional metrics $d_e$ (*no directions*). We provide IGGM results only for $N = 16$ as IGGM fails to run for $N = 128$ due to method-intrinsic memory limitations (Appendix A).

| Method | $N = 16$ | | | | $N = 128$ | | | |
|---|---|---|---|---|---|---|---|---|
| | GCD | Heat | Wave | Portrait | GCD | Heat | Wave | Portrait |
| *Parametric Stochastic Ensembles* | | | | | | | | |
| ER-mix | 4.410 | 1.767 | 6.001 | 0.634 | 5.868 | 0.068 | 2.646 | 0.811 |
| Random Graph Generators | 4.145 | 1.805 | 5.462 | 0.619 | 4.962 | 0.905 | 5.202 | 0.840 |
| *Combinatorial Search and Generative Bootstrapping* | | | | | | | | |
| ER→Genetic | 7.218 | 5.277 | 10.629 | 0.848 | 7.255 | 2.226 | 6.154 | 0.897 |
| ER→Genetic→Local Optimization | 7.344 | **5.279** | 10.754 | 0.863 | 7.395 | 1.235 | 5.878 | 0.929 |
| Greedy→Genetic | 7.292 | 5.278 | 10.727 | 0.853 | 7.760 | 5.434 | **10.974** | 0.959 |
| Greedy→Genetic→Local Optimization | **7.718** | 5.278 | **10.765** | 0.865 | 7.887 | 5.448 | 10.915 | 0.962 |
| Greedy | 6.747 | 4.686 | 9.816 | 0.835 | 7.360 | 2.854 | 8.151 | 0.941 |
| Greedy→Local Optimization | 7.486 | 5.277 | 10.671 | 0.858 | 7.781 | 2.972 | 8.325 | 0.955 |
| IGGM | 6.465 | 5.227 | 9.929 | 0.872 | — | — | — | — |
| *Ours* | | | | | | | | |
| Neural Graph Dispersion (NGD) | 7.621 | 5.277 | 10.634 | **0.885** | **7.981** | **5.546** | 10.674 | **0.969** |
| *NGD (no ensemble)* | 6.100 | 4.778 | 10.140 | 0.806 | 7.229 | 5.545 | 10.448 | 0.929 |
| *NGD (no directions)* | 6.242 | 5.275 | 10.257 | 0.823 | 7.283 | **5.546** | 10.565 | 0.957 |

## 4. Experiments

Our empirical study evaluates the efficacy and scalability of neural graph dispersion. We compare against prior baselines (Velikonivtsev et al., 2024) at small graph sizes under standard evaluation metrics $d_{GC}$, $d_{heat}$, $d_{wave}$, and $d_{PD}$ (Section 2.2). We then study behavior as the number of vertices $N$ and target set cardinality $k$ increase. Finally, we ablate key design choices and evaluate NGD generated graphs on two representative downstream tasks.

Baselines reported in Table 1 are drawn from and evaluated under comparable parameter settings as prior work (Section 2.4). Following standard practice, these baselines may be composed by using the output of one method to initialize the seed population of another; such compositions are indicated using the → notation. All methods are evaluated using an end-to-end candidate pool budget of size $M = 100,000$ to ensure fair comparison. Candidate pool budget is split evenly in composed methods. Implementation details and complexity analysis are provided in Appendix A. Additional experimental details are provided in Appendices D.

### 4.1. Small-Scale Experiments

We first evaluate neural graph dispersion in the small-graph regime, where prior graph-level dispersion methods are computationally feasible. As discussed in Section 2.4, existing approaches fall into three broad classes: parametric stochastic graph ensembles, combinatorial search methods, and generative bootstrapping pipelines. While the latter two achieve strong diversity at small scales, they rely on exhaustive, serial procedures that prior work acknowledges limits scalability (Velikonivtsev et al., 2024). Small-scale experiments therefore provide a controlled setting to assess diversity quality independently of scalability considerations.

Table 1 reports results for graph sizes $N = 16$ and $N = 128$. Across all metrics and graph sizes, neural graph dispersion is consistently competitive with the strongest baselines from Velikonivtsev et al. (2024). In particular, our approach attains near-best performance across multiple evaluation metrics, *despite not being trained to optimize those metrics*.

At small $N$, this distinction matters: combinatorial and bootstrapping pipelines can serially explore a non-negligible neighborhood of $S_N$ and thus may obtain modest gains on individual metrics (e.g., $d_{wave}$) by explicitly optimizing selection under that distance. In contrast, parametric stochastic ensembles consistently yield lower diversity scores, reflecting concentration near modes of their underlying distributions even when parameters are carefully tuned.

Overall, these results show that direct trajectory-based optimization in a surrogate pseudometric space can match the performance of serial search at small scales, while avoiding the algorithmic structures that restrict scalability. They further suggest that the learned surrogate geometry induces a general notion of structural diversity, which can be efficiently refined under bespoke evaluation distances through lightweight selection. We illustrate this structural variety

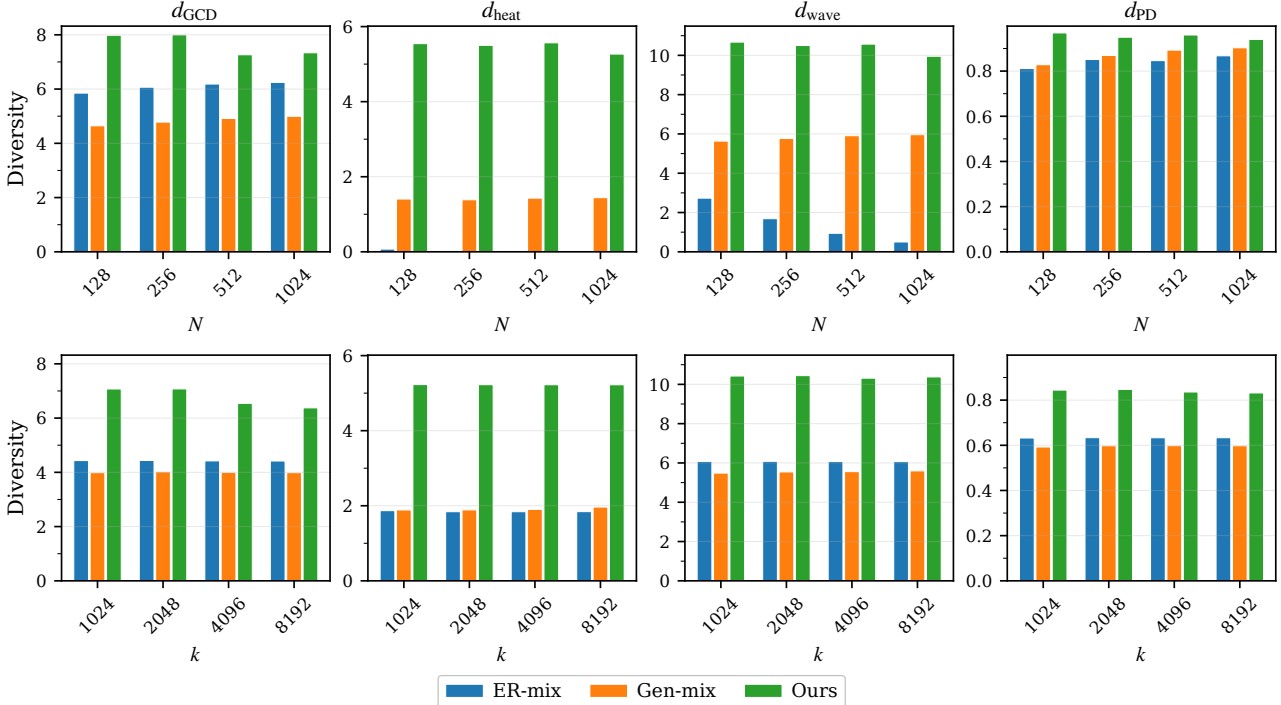

*Figure 2.* **Graph dispersion at large graph sizes and set cardinalities.** Top row: diversity as a function of vertices $N$ (with $k = 100$). Bottom row: diversity as a function of target set cardinality $k$ (with $N = 16$). Columns correspond to metrics $d_{\mathrm{GC}}$, $d_{\mathrm{heat}}$, $d_{\mathrm{wave}}$, and $d_{\mathrm{PD}}$ (higher is better). We compare neural graph dispersion to parametric stochastic ensembles ER-mix and Gen-mix. Combinatorial search and generative bootstrapping methods are omitted, as their serial procedures do not scale to this regime. Neural graph dispersion maintains higher structural diversity across metrics as $N$ and $k$ increase, while parametric ensembles exhibit saturation or degradation at scale.

qualitatively in Appendix E, which visualizes representative $N = 16$ graphs drawn from each evaluation distance. These observations motivate the scaling experiments presented next.

### 4.2. Scaling Experiments

We next study dispersion in regimes where the number of vertices $N$ or the target set cardinality $k$ exceeds those considered in prior work. As discussed throughout the manuscript, combinatorial search and generative bootstrapping methods are intractable in this regime. We therefore restrict comparisons to parametric stochastic ensembles ER-Mix and Gen-Mix (Section 2.4) that remain feasible at scale.

Figure 2 reports diversity as a function of $N$ and $k$. Across all metrics, neural graph dispersion maintains high structural diversity as $N$ increases, while ensemble baselines saturate or degrade. Similarly, diversity remains stable as $k$ grows, indicating that increasing the output set size does not dilute structural coverage. These results show that neural graph dispersion enables graph-level dispersion at values of $N$ and $k$ that are an order of magnitude larger than those accessible to prior iterative approaches while preserving high diversity across a range of structurally distinct metrics.

### 4.3. Ablation Studies

We conduct ablations to assess the contributions of two key components of neural graph dispersion: the generator ensemble and directional surrogate metrics. Results are reported in Table 1. Removing the ensemble and instead optimizing a single generator under the full feature-space distance consistently reduces diversity, indicating that complementary projection directions are critical for broad structural coverage. Replacing directional metrics with a single scalar projection of $\phi(G)$ likewise degrades performance, as collapsing high-dimensional structure into one direction weakens the dispersion signal. These results support the use of ensembles and multi-dimensional directional metrics.

### 4.4. Downstream Evaluation

We evaluate whether NGD's resultant metric separation corresponds to functionally distinct graphs that induce different behavior in downstream tasks. To do so, we stress-test two downstream targets: a learned graph-level predictor (GNN regression) and a classical combinatorial heuristic (greedy graph coloring). Appendix D contains full experimental details and results. In both cases, the evaluation criterion is not diversity itself but the ability to expose regimes that standard graph ensembles fail to reach.

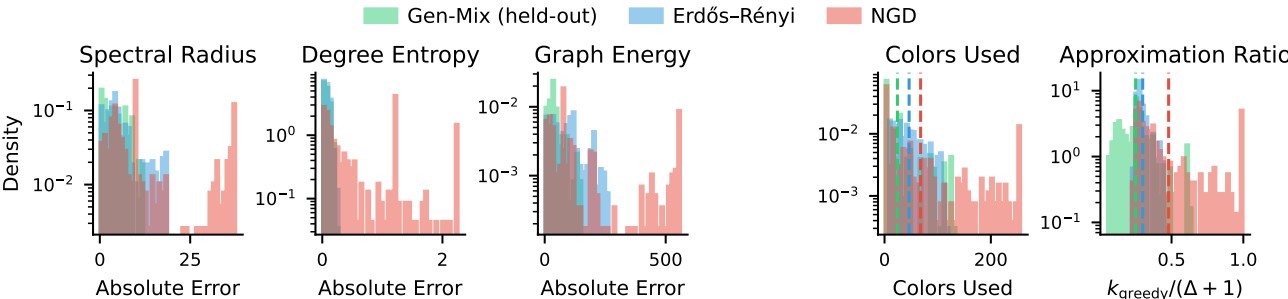

*Figure 3.* **Downstream stress tests using NGD-generated graphs.** Left three panels: distribution of per-graph absolute errors for GIN regressors trained on generator-mix graphs ($N = 256$) and evaluated on held-out generator-mix, Erdős–Rényi, and NGD test sets. Right two panels: greedy graph coloring (largest-first) applied to the same three ensembles, showing the distribution of colors used and the approximation ratio $k_{\text{greedy}}/(\Delta + 1)$. Dashed lines indicate ensemble means. NGD graphs exhibit heavy right tails in all panels, indicating structural regimes where both learned models and classical heuristics degrade, absent under standard graph ensembles.

**Graph Neural Network stress test.** We train independent 3-layer Graph Isomorphism Network (GIN) (Xu et al., 2019b) regressors on 5,000 generator-mix graphs ($N = 256$) for each of six regression targets spanning various graph properties, then evaluate on held-out generator-mix, Erdős–Rényi, and NGD-generated test sets. Table 2 reports mean absolute error (MAE) averaged over five trials. Across all targets, NGD graphs yield $3 - 11\times$ higher MAE than held-out generator-mix graphs. The per-graph error distributions (Figure 3, left three panels) reveal that this increase is driven by heavy right tails absent under both generator-mix and ER test sets: individual graphs on which the GNN's predictive capacity degrades sharply. These failures are consistent across targets and random seeds, indicating systematic coverage of structural regimes outside the training distribution rather than isolated outliers. Appendix D reports distributional results for all tested regression targets.

**Greedy graph coloring stress test.** We apply greedy graph coloring under two standard vertex orderings (largest-first and smallest-last) to the same three ensembles, measuring the approximation ratio $k_{\text{greedy}}/(\Delta+1)$, where $\Delta$ is the maximum degree and $\Delta + 1$ is the trivial upper bound from Brook's theorem (Brooks, 1941). Values near 1.0 indicate worst-case performance. NGD graphs yield a mean approximation ratio of $0.48$ versus $0.25$ for generator-mix under largest-first ordering, and the distribution extends to 1.0 (Figure 3, right two panels), a regime in which greedy coloring assigns a unique color to nearly every vertex. This regime is entirely absent under ER and generator-mix. Moreover, we find these results are consistent across both vertex orderings (See results in Table 5 in Appendix D).

Both experiments show that structural coverage produced by NGD extends beyond metric separation: NGD-generated graphs occupy regimes inaccessible to classical generators, exposing failure modes that narrow benchmarks conceal.

*Table 2.* **GNN stress test results** (mean MAE over 5 trials). A 3-layer GIN regressor is trained on 5,000 generator-mix graphs ($N = 256$) per target and evaluated on held-out generator-mix (500), Erdős–Rényi (500), and NGD-diverse (400) test sets. MAE is reported in original units; the highest MAE per row is bolded. See Appendix D for standard deviations.

| Target | Gen-Mix | ER | NGD |
|---|---|---|---|
| Spectral Radius | 3.2956 | 5.4802 | **10.5067** |
| Num. Comp. (norm) | 0.0092 | 0.0032 | **0.0497** |
| Graph Energy | 39.3451 | 96.1887 | **182.3352** |
| Global Efficiency | 0.0144 | 0.0163 | **0.0449** |
| Degree Entropy | 0.0674 | 0.0788 | **0.7659** |
| Avg. Neighbor Degree | 3.2146 | 5.2649 | **10.8418** |

## 5. Future Work

While this work focuses on graph-structured data, the proposed neural framework introduces minimal graph-specific inductive bias. An interesting direction for future work is to extend neural dispersion to other combinatorial domains, such as sets of permutations, matchings, or set systems, by designing appropriate continuous feature maps. Additionally, alternative dispersive potentials or generator architectures are briefly discussed in Appendix D and can be further studied to improve and extend our formulation. Finally, while Proposition 3.2 characterizes initialization as a latent inhomogeneous Erdős–Rényi family, optimization trajectories escape this family by construction. Theoretical characterization of graphs along or at convergence of these trajectories requires further analysis.

## 6. Conclusion

We have introduced *neural graph dispersion*, the first direct neural framework for generating structurally diverse graphs. Our approach achieves competitive diversity compared to brute-force methods at small scales and enables dispersion at order-of-magnitude larger node counts and set sizes.

## Acknowledgments

This research used resources of the Argonne Leadership Computing Facility, a U.S. Department of Energy (DOE) Office of Science user facility at Argonne National Laboratory and is based on research supported by the U.S. DOE Office of Science-Advanced Scientific Computing Research Program, under Contract No. DE-AC02-06CH11357. Additional funding support comes from the National Science Foundation (CCF-2119184 CNS-2313190 CCF-1822949 CNS-1956180). RW gratefully acknowledges the support of NSF DMS-2023109, DOE DE-SC0022232, the NSF-Simons National Institute for Theory and Mathematics in Biology (NITMB) through NSF (DMS-2235451) and Simons Foundation (MP-TMPS-00005320), and the Margot and Tom Pritzker Science Foundation.

## Impact Statement

This paper presents work whose goal is to advance the field of Machine Learning. There are many potential societal consequences of our work, none which we feel must be specifically highlighted here.

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

# A. Prior Work in Detail

## A.1. Overview of Graph Distance Constructions

We briefly expand on graph distance constructions beyond the descriptor-based metrics used for evaluation in the main body. Common approaches can be broadly classified into three categories. *Node-matching–based distances* define $d(G, G')$ via explicit (Gao et al., 2010; Riesen & Bunke, 2009; Bunke & Shearer, 1998) or relaxed (Xu et al., 2019a; Titouan et al., 2019; Zhou & De la Torre, 2016; Torresani et al., 2008) alignments between vertices, typically by optimizing over permutations or couplings between adjacency matrices. *Graphon-based methods* (Lovász, 2012; Sabanayagam et al., 2022; Mukherjee et al., 2017) lift finite graphs to limiting objects and compare them up to measure-preserving transformations, providing strong asymptotic guarantees but relying on dense-graph or large-$N$ assumptions. *Descriptor-based metrics* (Vishwanathan et al., 2010) define a Euclidean mapping $\phi(G) : S_N \to \mathbb{R}^q$ and measure distance via standard norms, $d(G, G') = \|\phi(G) - \phi(G')\|$. Accordingly, descriptor-based distances form the basis of the evaluation metrics used throughout our experiments.

## A.2. Formal Definitions of Evaluation Metrics

We briefly recall the standard definitions of the evaluation metrics used in our experiments. All metrics are isomorphism-invariant and are applied to simple, undirected graphs with $N$ vertices. We follow the original formulations in the respective references; this subsection is included for completeness and to fix notation.

**Graphlet Correlation Distance (GCD)** (Yaveroğlu et al., 2014). Graphlet Correlation Distance compares graphs via correlations between local graphlet orbit counts. Given a graph $G$, let $X_G \in \mathbb{R}^{n \times o}$ denote the matrix of node-level graphlet orbit counts up to graphlets of size four, where $o$ is the number of considered orbits. Following standard practice, $X_G$ is computed using ORCA (Hočevar & Demšar, 2014) and restricted to a fixed subset and ordering of orbits. The *graphlet correlation matrix* (GCM) of $G$ is then defined as $\text{GCM}(G) = \text{corr}_{\text{Spearman}}(X_G)$, where the correlation is computed between orbit count vectors across nodes. The Graphlet Correlation Distance between graphs $G$ and $G'$ is given by the Euclidean distance between the upper-triangular entries of their correlation matrices: $d_{\text{GC}}(G, G') = \left\| \text{vec}_\triangle\big(\text{GCM}(G)\big) - \text{vec}_\triangle\big(\text{GCM}(G')\big) \right\|_2$, where $\text{vec}_\triangle(A)$ denotes the vectorized upper-triangular of a matrix $A$.

**NetLSD Distances** (Tsitsulin et al., 2018). NetLSD compares graphs via spectral trace signatures derived from the normalized graph Laplacian. Given a graph $G$, let $\{\lambda_i\}_{i=1}^N$ denote the eigenvalues of the normalized Laplacian of $G$. The NetLSD *heat signature* is defined as $h_G(t) = \sum_{i=1}^N e^{-t\lambda_i}$, evaluated over a fixed grid of time scales $t$. The *wave signature* is defined as $w_G(t) = \sum_{i=1}^N \cos(t\lambda_i)$, evaluated over a fixed frequency grid. Both signatures yield graph-level vectors invariant to node relabeling and graph size. The corresponding distances between graphs $G$ and $G'$ are defined as Euclidean distances between signatures: $d_{\text{heat}}(G, G') = \|h_G - h_{G'}\|_2$, $\qquad d_{\text{wave}}(G, G') = \|w_G - w_{G'}\|_2$.

**Portrait Divergence (PD)** (Bagrow & Bollt, 2019). Portrait Divergence compares graphs via their network portraits, which encode the joint distribution of shortest-path distances and node degrees. Given a graph $G$, let $P_G$ be the portrait with $P_G(k, \ell)$ denoting the fraction of nodes that have exactly $\ell$ neighbors at graph distance $k$. The portrait distance between graphs $G$ and $G'$ is defined as the Jensen–Shannon divergence between the corresponding portraits: $d_{\text{PD}}(G, G') = \text{JSD}(P_G \| P_{G'})$, which yields a symmetric, finite divergence that captures differences in global distance structure.

## A.3. Parametric Stochastic Ensemble Graph Generators

Parametric stochastic ensembles are common baselines for graph dispersion using well-established graph generator models. Following on prior work (Velikonivtsev et al., 2024), we adopt two such ensembles: *ER-mix* and *Generator-mix*. The former entails only Erdős–Rényi ensembles whereas the latter includes a mixture of standard generators discussed below. Across both methods, we maintain the same generator parameters as Velikonivtsev et al. (2024), but alter the node count ($N$) and other related parameters when necessary. Each ensemble is sampled an equal number of times to obtain a candidate pool $\mathcal{C}$.

**Erdős–Rényi** (Erdős & Rényi, 1959). These models sample each potential graph edge with a fixed probability.

**Preferential Attachment** (Barabási & Albert, 1999). These generators iteratively add new nodes to the graph and connect the new nodes to prior ones stochastically. Prior nodes with higher degrees are more likely to be connected to new ones.

**Holme-Kim** (Holme & Kim, 2002). A modification to preferential attachment generators, these ensembles decide with a fixed probability between adding triangle forming or stochastic, degree-based edges.

**Power-Law Degree model** (Aiello et al., 2001). Random graphs are generated by connecting nodes with probabilities derived from a power-law distribution.

**Random Geometric Graphs** (Penrose, 2003). These ensembles stochastically places graph nodes in a unit cube in $\mathbb{R}^d$ and add an edge between two nodes if the distance between them is less than a parametric $r$.

**Random Regular Graphs.** Regular, $d$-degree graphs are generated at random for a variety of $d$.

**Stochastic Block Model** (Holland et al., 1983). Nodes are divided among a number of fixed-size blocks. Intra-, and inter-block edges are then added with fixed probabilities $p$ and $q$.

### A.4. Combinatorial Search and Generative Bootstrapping Baselines

We provide a brief overview of prior combinatorial search and generative bootstrapping baselines from our experimental evaluation. In general, all approaches generate undirected graphs without self-loops and with a fixed number of $N$ unlabeled vertices. Note that these baselines may be composed by using the output of one method as the pre-generated seed population for another (e.g., Greedy algorithm followed by the Genetic one). Except node count ($N$), all default parameters were taken from Velikonivtsev et al. (2024); Appendix D outlines the experimental setup.

Additionally, we analyze prior work's computational complexity under fixed node count $N$ and target set size cardinality of $k$ based on the following, metric-agnostic notation: let $d(G, G') : S_N \times S_N \to \mathbb{R}_{\geq 0}$ be an arbitrary metric with $d(G, G') := \zeta(\phi(G), \phi(G'))$, where $\phi(G) : S_N \to \mathbb{R}^q$ is a feature map and $\zeta(\cdot, \cdot) : \mathbb{R}^q \times \mathbb{R}^q \to \mathbb{R}_{\geq 0}$ is the distance between two feature maps. We denote the complexity of $\phi$ and $\zeta$ as $\mathcal{O}(\phi)$ and $\mathcal{O}(\zeta)$.

**Greedy Algorithm.** Given a candidate set $\mathcal{C}$ (where $M := |\mathcal{C}|$ and $M > k$) generated from the Generator-mix stochastic ensemble, the Greedy algorithm applies a standard greedy max-sum selection procedure on $\mathcal{C}$ with respect to an evaluation metric $d_{\text{eval}}$ to produce a set $\mathcal{G} \subset \mathcal{C}$ with $|\mathcal{G}| = k$. To do so, the algorithm first pre-processes $\phi(G)$ for all $G \in \mathcal{C}$ and starting with $\mathcal{G} = \emptyset$, selects a random graph in $\mathcal{C}$ be the first element in $\mathcal{G}$. The algorithm then repeats the following for $k - 1$ iteration: the graph $G \subset \mathcal{C}$ that is most dissimilar to current elements in $\mathcal{G}$ (based on Equation 1 and using $\zeta(\cdot, \cdot)$) is appended to $\mathcal{G}$. As shown by Velikonivtsev et al. (2024), this procedure requires $\mathcal{O}(\phi M)$ for pre-processing and $\mathcal{O}(\zeta M)$ calculations per iteration. Hence, the total computations complexity is $\mathcal{O}((\phi + \zeta k)M)$.

**Genetic Algorithm.** Given a pre-generated seed population $\mathcal{G}$ (from ER-mix or output of another algorithm and where $k := |\mathcal{G}|$), this method explores $S_N$ via crossover and mutation of elements in $\mathcal{G}$, optimizing $\mathcal{G}$ with respect to evaluation metric $d_{\text{eval}}$. At each iteration, $G_1, G_2 \subset \mathcal{G}$ are chosen randomly as parents and via mutation and crossover (Velikonivtsev et al., 2024), generate a potential candidate graph $G'$. The algorithm then attempts to replace each element of $\mathcal{G}$ with $G'$, calculating the approximate change in diversity caused by replacement and successfully updates $\mathcal{G}$ if the replacement improves diversity. Irrespective of replacement success, the algorithm is limited to $M$ total graph generations (i.e., iterations). Hence this approach requires $\mathcal{O}(\phi + \zeta k)$ complexity per generated graph, with a total of $\mathcal{O}((\phi + \zeta k)M)$ across the entire process (Velikonivtsev et al., 2024).

**Local Optimization (LocalOpt) Algorithm.** Given a pre-generated seed population $\mathcal{G}$ (from ER-mix or output of another algorithm and where $k := |\mathcal{G}|$), this algorithm explores $S_N$ via edge perturbations of elements in $\mathcal{G}$, optimizing $\mathcal{G}$ with respect to evaluation metric $d_{\text{eval}}$. At each iteration, a random $G \subset \mathcal{G}$ is chosen and its edges are randomly perturbed to generate a candidate graph $G'$. The algorithm then replace $G$ with $G'$ if such replacements leads to improved diversity, attempting the random graph generation process for $M$ iterations. Thus, the per iteration complexity is $\mathcal{O}(\phi + \zeta k)$, for a total of $\mathcal{O}((\phi + \zeta k)M)$ across the entire process (Velikonivtsev et al., 2024).

**Iterative Graph Generative Modeling (IGGM).** As described, this neural generative algorithm operates via a $T$ step iterative process. At each iteration $t$, the algorithm fits a new neural generative model to a candidate pool $\mathcal{C}_t \subset S_N$, where $R := |\mathcal{C}_t|$ (the first iteration is seeded with graphs generated from Generator-mix). Once complete, the neural generator

*Table 3.* Neural graph dispersion uses less memory and can be easily scaled with respect to $N$ with multi-GPU training. IGGM, in contrast, is unable to run with $N \geq 128$ on our experimental setup due to lack of GPU memory and multi-GPU support. With the exception of $N$, no configuration parameters were altered.

| Neural Generator | $N = 16$ | $N = 32$ | $N = 64$ | $N = 128$ | $N = 256$ | $N = 512$ |
|---|---|---|---|---|---|---|
| IGGM | 1,565 | 5,701 | 21,554 | — | — | — |
| Neural Graph Dispersion (ours) | 1,880 | 3,432 | 9,913 | 35,840 | 133,027 | 517,986 |

is used to sample a graph pool $\mathcal{G}_t$ — where $P := |\mathcal{G}_t|$ — and the Greedy algorithm is used to choose $R$ diverse graphs with respect to $d_{\text{eval}}$ that will form $\mathcal{C}_{t+1}$. The last iteration chooses $k$ diverse graphs to form the final, resultant graph set. Velikonivtsev et al. (2024) utilize the Discrete Denoising Diffusion model DiGress (Vignac et al., 2023) with default parameters as the underlying generative model for IGGM.

**Generated Graph Budget.** Adopting prior work's approach, we limit all approaches to the same number of $\mathcal{O}(\phi + \zeta k)$ computations. In our experiments, we set $M := 100{,}000$ such computations as the end-to-end budget for all experiments. For composed methods, the budget is split evenly between different algorithm (e.g., in the case of the Greedy algorithm followed by the Genetic algorithm, each algorithm is allocated a budget of 50,000). For IGGM, given that metric values are only calculated during the Greedy portion of the algorithm, we set $T \cdot P := 100{,}000$ as the end-to-end budget, with $T := 10, P := 10{,}000$, and $R := 1000$.

**Scalability Limitations.** We briefly discuss the algorithmic components of prior work that limit scalability under the experimental setup outlined in Appendix D. In general, *neural graph dispersion scales better than prior baselines at larger values of $N$ without sacrificing metric performance*. For instance, we found that increasing the node count ($N$) from 16 to 128 led to an average $5.17\times$ increase in runtime for Genetic algorithm across all evaluation metrics but only an average $2.6\times$ increase in neural graph dispersion's runtime. For $N \geq 256$, all methods consisting of Genetic and Local Optimization Algorithm exceed the 12 hour wall-clock budget (Appendix D) due to their serial nature. For the greedy algorithm, the generation of the required candidate set $\mathcal{C}$ with $N \geq 256$ exceeded the wall-clock budget. Neural graph dispersion, on the other hand, completed all experiments within the allocated wall-clock budget.

More importantly, the Genetic and Local Optimization algorithms are serial by nature, limiting their scalability. While the inter-step computations for updating $\mathcal{G}$ for either approach may be conducted in parallel, the overall algorithm limits graph generation to one graph at a time, effectively enforcing a serial operating regime. In contrast, the Greedy algorithm (both as a baseline and as component of neural graph dispersion) is scalable by design. First, pre-processing of $\phi$ calculations are independent of each other and thus, parallelizeable. Second, the greedy max-sum procedure and $\zeta$ calculations for each step benefit from vectorized operations amenable to GPU execution, vastly improving the runtime. Given that typically $M >> k$, the Greedy Algorithm requires far fewer serial iterations than either Genetic or Local Optimization algorithms. These optimizations result in an average 3-12 $\times$ speedup for the Greedy algorithm (and by extension, neural graph dispersion).

Next, IGGM, the only neural generative baseline, is unscalable due to both memory and runtime constraints. For instance, IGGM with $N = 16$ takes an order of magnitude longer than our neural generative dispersion algorithm with $N = 128$. Moreover, prior implementation of IGGM does not support multi-GPU training and as shown in Table 3, its single-GPU setup fails to run for graphs with $N \geq 128$. In contrast, our neural generative model and candidate selection require less memory than IGGM and supports multi-gpu training (e.g., by placing each generator in a different GPU).

## B. Extended Details on Neural Graph Dispersion

### B.1. Extended Definitions and Implementation Details

We briefly provide additional formal definitions and implementation-level clarifications for the feature maps and objectives used throughout the paper. All constructions described here are deterministic given a fixed random seed and do not introduce additional modeling assumptions beyond those stated in the main manuscript.

**Spectral Moment Features.** Given a power $r$ and a graph $G$ with adjacency matrix $A \in \{0, 1\}^{N \times N}$ and diagonal degree matrix $D \in \mathbb{Z}_{\geq 0}^{N \times N}$, we define *adjacency spectral moments* via $m_r(A) := \text{tr}(A^r) = \sum_{i=1}^{N} \lambda_i^r$, where $\{\lambda_i\}$ are the eigenvalues of $A$. Analogously, let $L = D - A$ denote the (unnormalized) graph Laplacian; we define *Laplacian moments*

$m_r(L) := \mathrm{tr}(L^r)$. We avoid explicit eigendecomposition by computing these features directly via matrix powers.

Rather than using raw moments, we construct a set of normalized *moment ratios* designed to be stable and comparable across graphs of different sizes. Typical examples include $\frac{m_3(A)}{m_2(A)}, \frac{m_4(A)}{m_2(A)}, \frac{m_6(A)}{m_4(A)}$, as well as analogous ratios involving Laplacian moments and cross-ratios between adjacency and Laplacian moments. These ratios capture structural properties such as clustering, regularity, and long-range walk behavior while remaining invariant to node relabeling.

**Stabilization and Normalization.** To ensure numerical stability, all ratios are computed using $\frac{x}{|y|+\varepsilon}$, with a fixed $\varepsilon > 0$, and are masked when denominators are near zero. Additionally, feature values are normalized using graph-size-dependent scaling factors (e.g., powers of $N$) so that features lie in a comparable numeric range across different $N$. This normalization avoids degenerate behavior for sparse or near-empty graphs and improves optimization stability.

**Ratio Configuration and Selection.** Let $\phi(G) \in \mathbb{R}^q$ denote the full vector of enabled spectral moment ratios, where $q$ depends on the selected subset of ratios. The enabled feature set is controlled by a binary feature mask, allowing individual ratios or entire families of ratios to be included or excluded without changing the underlying computation. All spectral features and ratios are computed from symmetric, loop-free adjacency matrices.

**Random Linear Projections.** To obtain a low-dimensional embedding suitable for efficient optimization, we apply a fixed random linear projection to the feature vector. For each ensemble index $e$, we sample projection directions $U_e \in \mathbb{R}^{q_e \times q}$, with rows normalized to unit norm, and define the projected representation as $U_e(\phi(G) - \phi(G')) := R_e(\phi(G) - \phi(G')) \in \mathbb{R}^{q_e}$. The projection dimension $q_e$ is small (typically $q_e \leq 4$), and all projection matrices are fixed throughout training. This construction preserves relative distances between feature vectors in expectation while enabling multiple ensembles to explore distinct directions in feature space.

**Pairwise Distances.** Given a minibatch of graphs ($\{G_i\}_{i=1}^b$), pairwise distances are computed using the Euclidean metric for all $G_i, G_j$ with $i \neq j$, $d(G_i, G_j) := \|U_e(\phi(G_i) - \phi(G_j))\|_2$. These distances are used both during training (to define repulsive objectives) and during postprocessing for candidate selection.

## B.2. Candidate Selection Procedure

We describe the candidate selection procedure used to construct a diverse set of $k$ graphs from a collection of learned probability matrices. Our two-stage procedure allows the candidate pool $\mathcal{C}$ to cover a broad range of structural regimes while keeping both sampling and distance evaluation computationally tractable. The procedure is metric-agnostic and applies to any evaluation distance $d_{\mathrm{eval}}$ described in Appendix A.

**Setup.** Let $\psi = \{\widetilde{A}_h\}_{h=1}^H$ denote a collection of symmetric edge-probability matrices, with $\widetilde{A}_h \in [0,1]^{N \times N}$. Sampling from $\widetilde{A}_i$ produces a random graph $G$ with adjacency matrix $A \sim \widetilde{A}_h$ by independently drawing edges according to Bernoulli trials, optionally modified by a temperature parameter. The goal is to select a final set $\mathcal{G}$ of $k$ graphs that maximizes diversity under $d_{\mathrm{eval}}$ while keeping the total sampling budget bounded. We begin with an empty candidate pool $\mathcal{C}$.

**Phase I: Matrix-level diversity discovery.** In the first phase, we identify a diverse subset of probability matrices. For each matrix $\widetilde{A}_h$, we sample a single representative graph $G_h$ with $A_h \sim \widetilde{A}_h$ at unit temperature and add them to the candidate pool $\mathcal{C}$. Each representative graph is then embedded using the chosen $d_{\mathrm{eval}}$, yielding either an explicit embedding vector (e.g., NetLSD or GCD) or an implicit representation (e.g., network portraits).

We then apply a greedy max-sum selection procedure to the set $\{G_h\}_{h=1}^H$, selecting the indices of the $L \ll H$ most diverse matrices. Initialization is performed either randomly or using a heuristic that selects an approximately maximally dissimilar pair. For metrics whose evaluation is expensive (e.g., Portrait Divergence), distances are computed incrementally during the greedy procedure rather than forming the full pairwise distance matrix.

**Phase II: Focused graph sampling.** In the second phase, sampling is restricted to the selected subset of probability matrices. Until $|\mathcal{C}| := M$, we draw multiple graph samples from each retained matrix $\widetilde{A}_h$ using either discrete or continuous temperature sampling, as well as an optional deterministic sampling mode based on expected edge count. This process completes the production of a candidate pool $\mathcal{C}$ whose size is controlled by the fixed sampling budget.

**Final selection.** Given the finalized pool $\mathcal{C}$, we perform a final greedy max-sum selection under $d_{\text{eval}}$ to obtain the final set $\mathcal{G} \subset \mathcal{C}$ with $|\mathcal{G}| := k$. The greedy objective can be either the sum of pairwise distances or an energy-based objective proportional to the inverse distance. As in Phase I, distance computations are performed incrementally when applicable.

**Training and Selection Graph Budgets** Following on standards outlined in Appendix A, we set a maximum budget of $M = 100{,}000$ for the final candidate pool size. During training, we collect $H = 10{,}000$ relaxed adjacency matrices, which are narrowed down to $L = 1000$ after Phase I of the candidate selection. Finally, Phase II samples each retained $\widetilde{A}_h$ 90 times to generate 90,000 additional graphs, resulting in a candidate pool of 100,000 graphs.

## C. Proofs

**Proposition 3.1** (Stability and repulsion of Coulomb objectives). *Assume $d(\cdot, \cdot)$ is differentiable on the relaxed graph domain and that its gradient is bounded on compact sets. If there exists $\delta > 0$ such that $d(G_i, G_j) \geq \delta$ for all $i \neq j$, then the Coulomb loss $\mathcal{L}(\mathcal{G}; d)$ and its gradients with respect to the generated graphs are finite and uniformly bounded. Conversely, as $d(G_i, G_j) \to 0$ for any pair $(i, j)$, the repulsive gradient $|\nabla_{G_i} \ell_{ij}|$ grows as $(d(G_i, G_j) + \varepsilon)^{-(\gamma+1)}$ grows, approaching $\infty$ when $\varepsilon = 0$ provided $|\nabla_{G_i} d(G_i, G_j)|$ remains bounded away from zero.*

*Proof.* Let

$$\mathcal{L}(\mathcal{G}; d) = \sum_{i \neq j} \ell_{ij}, \qquad \ell_{ij} := \big(d(G_i, G_j) + \varepsilon\big)^{-\gamma}.$$

*Bounded loss under separation.* If $d(G_i, G_j) \geq \delta$ for all $i \neq j$, then for every ordered pair $(i, j)$,

$$\ell_{ij} \leq (\delta + \varepsilon)^{-\gamma},$$

and therefore

$$\mathcal{L}(\mathcal{G}; d) \leq \sum_{i \neq j} (\delta + \varepsilon)^{-\gamma} = k(k-1)(\delta + \varepsilon)^{-\gamma},$$

so the loss is finite and uniformly bounded on the set of configurations with minimum pairwise distance at least $\delta$.

*Bounded gradients under separation.* Fix $i \in [k]$. By the chain rule,

$$\nabla_{G_i} \mathcal{L}(\mathcal{G}; d) = \sum_{j \neq i} \nabla_{G_i} \ell_{ij} + \sum_{j \neq i} \nabla_{G_i} \ell_{ji}.$$

For each $j \neq i$,

$$\nabla_{G_i} \ell_{ij} = -\gamma \big(d(G_i, G_j) + \varepsilon\big)^{-(\gamma+1)} \nabla_{G_i} d(G_i, G_j),$$

and similarly for $\nabla_{G_i} \ell_{ji}$. By assumption, $d(G_i, G_j) \geq \delta$ and $\|\nabla_{G_i} d(G_i, G_j)\| \leq L_d$ on the (compact) relaxed domain, hence

$$\|\nabla_{G_i} \ell_{ij}\| \leq \gamma(\delta + \varepsilon)^{-(\gamma+1)} L_d, \qquad \|\nabla_{G_i} \ell_{ji}\| \leq \gamma(\delta + \varepsilon)^{-(\gamma+1)} L_d.$$

Summing over $j \neq i$ and applying the triangle inequality gives

$$\|\nabla_{G_i} \mathcal{L}(\mathcal{G}; d)\| \leq 2(k-1)\gamma L_d (\delta + \varepsilon)^{-(\gamma+1)},$$

so the gradients are finite and uniformly bounded whenever all pairwise distances are bounded below by $\delta$.

*Strong repulsion near collisions.* For any fixed pair $(i, j)$, the same chain-rule expression yields

$$\|\nabla_{G_i} \ell_{ij}\| = \gamma \big(d(G_i, G_j) + \varepsilon\big)^{-(\gamma+1)} \|\nabla_{G_i} d(G_i, G_j)\|.$$

Thus, as $d(G_i, G_j) \to 0$, the magnitude of the repulsive force grows like $(d(G_i, G_j) + \varepsilon)^{-(\gamma+1)}$. In particular, in the idealized case $\varepsilon = 0$, if $\|\nabla_{G_i} d(G_i, G_j)\|$ does not vanish as the pair approaches collision, then $\|\nabla_{G_i} \ell_{ij}\| \to \infty$, yielding strong separation pressure on near-duplicate configurations. $\square$

**Proposition 3.2** (Initialization induces a restricted random graph family). *At random initialization, graphs generated by $f_\theta$ and sampled via independent Bernoulli draws from the predicted edge probabilities form a latent inhomogeneous Erdős–Rényi family: conditional on the latent noise $z$, edges are independent with probabilities given by $f_\theta(z)$.*

*Proof.* At initialization, the generator produces a matrix of edge probabilities $\widetilde{A} = f_\theta(z) \in [0,1]^{N \times N}$ for a sampled latent vector $z$. Discrete graphs are obtained by independently sampling each edge $A_{ij} \sim \text{Bernoulli}(\widetilde{A}_{ij})$ for $i < j$, with symmetry and zero diagonal enforced. Conditional on $z$, the resulting adjacency matrix therefore has independent edges with non-identical probabilities, which is precisely an inhomogeneous Erdős–Rényi model. Marginalizing over $z$ yields a latent-variable mixture of such models. □

**Proposition 3.3** (Isomorphism invariance of spectral features). *If $G \cong G'$ then $\phi(G) = \phi(G')$.*

*Proof.* If $G \cong G'$, then there exists a permutation matrix $P \in \{0,1\}^{N \times N}$ such that the adjacency matrices satisfy

$$A' = PAP^\top.$$

For any integer $r \geq 1$,

$$\text{tr}\big((A')^r\big) = \text{tr}\big((PAP^\top)^r\big) = \text{tr}\big(PA^rP^\top\big) = \text{tr}\big(A^r\big),$$

where we used $(PAP^\top)^r = PA^rP^\top$ (since $P^\top P = I$) and invariance of trace under cyclic permutations. Therefore every spectral-moment term appearing in $\phi$ (e.g., $\text{tr}(A^r)$) is identical for $G$ and $G'$. Moreover, any normalized feature constructed from these traces (such as the normalized moments $\mu_r$ in (3) and fixed ratios of traces) is also identical for $G$ and $G'$. Since $\phi(G)$ is obtained by concatenating a fixed collection of such trace-based features, we conclude $\phi(G) = \phi(G')$. □

**Proposition 3.4** (Completeness of spectral moments up to cospectrality). *Let $A$ be the adjacency matrix of a graph $G$. Then the moment sequence $(\text{tr}(A^1), \ldots, \text{tr}(A^N))$ uniquely determines the spectrum of $A$ (with multiplicity), and conversely. Thus, any graph feature map from which these moments can be recovered is complete up to cospectrality.*

*Proof.* Let $\lambda_1, \ldots, \lambda_N$ denote the eigenvalues of $A$ (counted with multiplicity). Since $A$ is symmetric,

$$\text{tr}(A^r) = \sum_{i=1}^{N} \lambda_i^r, \qquad r \geq 1.$$

Thus $(\text{tr}(A^1), \ldots, \text{tr}(A^N))$ is the sequence of power sums of the eigenvalues. By the Newton identities, the power sums up to order $N$ uniquely determine (and are uniquely determined by) the coefficients of the characteristic polynomial, and hence the spectrum of $A$. The claim follows. □

**Proposition 3.5** (Separation under random projections). *Let $x, y \in \mathbb{R}^q$ with $x \neq y$, and let $U \in \mathbb{R}^{q_e \times q}$ be a random matrix drawn from a continuous distribution. Then $Ux \neq Uy$ almost surely.*

*Proof.* Let $v := x - y \neq 0$. Then $Ux = Uy$ if and only if $Uv = 0$. The set of matrices $U \in \mathbb{R}^{q_e \times q}$ satisfying $Uv = 0$ is a proper linear subspace of $\mathbb{R}^{q_e \times q}$ and therefore has Lebesgue measure zero. Since $U$ is drawn from a continuous distribution, $\mathbb{P}(Uv = 0) = 0$, and hence $Ux \neq Uy$ almost surely. □

**Proposition 3.6** (Computational Complexity with respect to $N$ and $k$). *For fixed model architecture and optimization hyperparameters, the cost of a single gradient update in neural graph dispersion scales as $O(N^3)$, with no dependence on the target set cardinality $k$.*

*Proof.* Consider a single gradient update for one expert on a minibatch of $b$ relaxed graphs, where $b$, the maximum moment order $R$, projection dimension $q_e$, and the generator architecture are treated as fixed hyperparameters.

First, the generator outputs $b$ relaxed adjacency matrices in $[0,1]^{N \times N}$, which requires writing $\Theta(bN^2)$ entries and thus costs $O(N^2)$ with respect to $N$.

Next, the feature map computes spectral-moment features up to order $R$. For dense matrices, forming $\widetilde{A}^r$ (e.g., via repeated multiplication) costs $O(N^3)$ per power, and computing $\text{tr}(\widetilde{A}^r)$ adds $O(N)$. Repeating for $r = 2, \ldots, R$ across $b$ graphs yields $O(bRN^3) = O(N^3)$ since $b$ and $R$ are constant.

Finally, the repulsive loss evaluates all $\binom{b}{2} = O(1)$ pairwise distances in $\mathbb{R}^{q_e}$, costing $O(b^2 q_e) = O(1)$ with respect to $N$. Therefore, the total cost of one gradient update scales as $O(N^3)$ (with respect to $N$), and has no dependence on the target set cardinality $k$. $\qquad\square$

# D. Experimental Details

**Hardware and software tools.**    All neural experiments were implemented in Python using PyTorch (Paszke et al., 2019). To ensure reproducibility, we include a list of software dependencies in the supplementary material. GPU-dependent experiments were conducted using an AMD EPYC 20 7713 64-Core Processor running Red Hat Enterprise Linux 9.3 and a NVIDIA DGX A100 GPUs (40GB memory). For hyperparameter sweeps, a cluster of 8 such GPUs were used to run parallel experiments. CPU-only experiments, including combinatorial search baselines from prior work, were conducted on an AMD EPYC 9354 32-Core Processor running Ubuntu 22.04.5. Runtime measurements were recorded in the absence of other computationally heavy processes.

**Wall-Clock Budget Maximum.**    To ensure comparability of wall-clock budgets, we restrict evaluation to runs completing within 12 hours. Several serial baselines at larger graph sizes (e.g., $N \geq 256$) exceeded this budget per run and are therefore omitted in our evaluations.

**Unified Pipeline Overview.**    As described in the main manuscript, neural graph dispersion consists of two stages. First, an ensemble of $E$ neural generators is trained to optimize a surrogate dispersion objective in feature space. Second, a selection stage produces a final set of $k$ graphs under a specified evaluation distance. For large graphs, selection is performed in a memory-efficient manner by processing candidates in chunks.

## D.1. Neural Generator and Training Objective

**Generator Parameterization.**    For a fixed number of vertices $N$, each generator outputs a dense adjacency probability matrix $\widetilde{A} \in [0,1]^{N \times N}$. The matrix is parameterized by a multilayer perceptron (MLP) operating on latent noise vectors, followed by symmetrization and removal of self-loops:

$$\widetilde{A} \leftarrow \sigma\left(\tfrac{1}{2}(X + X^\top)\right), \qquad \widetilde{A}_{ii} \leftarrow 0.$$

We train an ensemble of $E$ such generators in parallel with independent parameters but vectorized computation. Each generator uses a fixed-depth MLP with ReLU activations and latent dimension 16.

**Straight-through Sampling.**    To propagate gradients through discrete graph samples, we use a multi-sample straight-through estimator (Bengio et al., 2013). Given $\widetilde{A}$, we sample Bernoulli graphs $A \sim \text{Bernoulli}(\widetilde{A})$ and use the straight-through estimator $A_{\text{STE}} = \widetilde{A} + (A - \widetilde{A})_{\text{stopgrad}}$ during backpropagation. Unless stated otherwise, we use 25 samples per forward pass.

**Surrogate Dispersion Objective.**    Let $\phi(\cdot)$ denote the surrogate feature map from Section 2.2, and let $d_e$ be the induced Euclidean distance in feature space. Each generator minimizes a Coulomb-style repulsive objective (Equation 2) over a $b$ sized batch of sampled graphs:

$$\mathcal{L} = \frac{1}{b(b-1)} \sum_{i \neq j} \frac{1}{(d_e(G_i, G_j) + \varepsilon)^\gamma},$$

where $\varepsilon > 0$ ensures numerical stability and $\gamma > 0$ controls repulsive strength. In all experiments reported in the main paper, we fix $\gamma = 0.2$.

**Potential Alternative Methods and Early Challenges.**    While our framework is agnostic to the choice of repulsive potential or generator architecture, alternative methods were not experimentally successful due to a variety of challenges. Addressing these challenges could potentially improve and extend our formulation in future work.

In particular, we evaluated alternative potentials such as mean pairwise distance, DPP-style objectives, and $k$-nearest-neighbor repulsion, but found them to exhibit training instabilities (e.g., mode collapse, unstable gradients, or oscillatory dynamics) or suboptimal performance (e.g., clumped graphs with limited diversity).

Regarding generators, we found that despite its simplicity, the MLP generator achieves state-of-the-art dispersion across tested metrics and at scale (Tables 6 and 7, Figure 2). GNN-based generators are problematic in our setting: graph generation from noise requires producing structure from scratch, whereas GNNs operate on an existing input graph. Transformer-based generators also performed suboptimally in our initial experiments.

### D.2. Surrogate Features and Random Directions

**Surrogate Features.** Surrogate features are constructed from normalized ratios of adjacency spectral moments $m_r(A) := \text{tr}(A^r)$, with $r \leq 6$, as formally defined in Appendix B. Across all reported experiments, we exclusively use adjacency-based moment ratios and disable all Laplacian and cross (adjacency–Laplacian) ratios.

Specifically, the feature set consists of the following moment ratios:

$$\frac{m_3(A)}{m_2(A)}, \quad \frac{m_4(A)}{m_2(A)}, \quad \frac{m_5(A)}{m_3(A)}, \quad \frac{m_6(A)}{m_4(A)}, \quad \frac{m_4(A)}{m_3(A)}, \quad \frac{m_6(A)}{m_2(A)},$$

together with additional higher-order ratios

$$\frac{m_5(A)}{m_2(A)}, \quad \frac{m_6(A)}{m_3(A)}, \quad \frac{m_5(A)}{m_4(A)},$$

and two clustering-sensitive normalizations,

$$\frac{m_3(A)^2}{m_6(A)}, \quad \frac{m_3(A)}{m_2(A)^{3/2}}.$$

All ratios are stabilized by small additive constants in the denominator and normalized to comparable numeric ranges, as described in Appendix B. These features capture local clustering, regularity, and long-range walk structure while remaining permutation-invariant and computationally stable.

**Random Directions.** To define directional surrogate distances, each ensemble member is assigned a set of random unit directions in feature space. For generator $e$, features are projected using a matrix $U_e \in \mathbb{R}^{q_e \times q}$ with $q_e \in \{1, 2, 3, 4\}$. All directions are generated deterministically from a fixed random seed and cached for reproducibility.

### D.3. Other Training and Selection Details

**Optimizer and Training Schedule.** All generators are trained using the Adam optimizer with a fixed learning rate. Training is run for a fixed number of iterations under a constant batch size. We do not use learning-rate decay or gradient clipping in the reported experiments.

**Relaxed Candidate Collection.** Generated adjacency probability matrices $\widetilde{A}_h$ — described in Appendix B — are periodically collected during training for downstream selection. The number of collected matrices is fixed to 10,000 across all $E$ ensemble members, resulting in approximately $10{,}000/E$ candidates per generator. We set the collection period $r = \max\left(1, \lfloor ET/10{,}000 \rfloor\right)$.

### D.4. Hyperparameter sweeps

**Sweep Protocol.** Each sweep trial trains an ensemble, performs selection, and logs diversity scores. For each experimental setting, we use the best configuration selected under the primary metric used in that experiment; best configurations are reported in the supplemental code base.

**Swept and Fixed Parameters.** For the main experiments, hyperparameters were selected via random search. We sweep the batch size uniformly over the range $[10, 100]$ and the number of ensemble members $E$ uniformly over the range $[1, 100]$. Generator architectures are varied by sweeping the MLP width over $\{256, 310, 384\}$ and depth over $\{3, 4, 5, 6, 7, 8\}$ hidden layers. The projection dimension used for directional surrogate distances is swept over $q_e \in \{1, 2, 3, 4, 5\}$.

In addition, we sweep over subsets of adjacency-derived surrogate features described in Appendix B.

*Table 4.* **GNN stress test results** (mean ± std over 5 trials). A 3-layer GIN regressor is trained on 5,000 generator-mix graphs ($N = 256$) per target and evaluated on held-out generator-mix (500), Erdős–Rényi (500), and NGD-diverse (400) test sets. MAE is reported in original units; the highest MAE per row is bolded.

| Target | Gen-Mix | ER | NGD |
|---|---|---|---|
| Spectral Radius | 3.2956 ± 0.7014 | 5.4802 ± 0.6663 | **10.5067 ± 2.7333** |
| Num. Components (norm) | 0.0092 ± 0.0047 | 0.0032 ± 0.0018 | **0.0497 ± 0.0295** |
| Graph Energy | 39.3451 ± 7.8122 | 96.1887 ± 10.0259 | **182.3352 ± 30.3486** |
| Global Efficiency | 0.0144 ± 0.0010 | 0.0163 ± 0.0050 | **0.0449 ± 0.0064** |
| Degree Entropy | 0.0674 ± 0.0045 | 0.0788 ± 0.0100 | **0.7659 ± 0.0430** |
| Avg. Neighbor Degree | 3.2146 ± 0.3971 | 5.2649 ± 1.3475 | **10.8418 ± 2.0506** |

Unless explicitly swept, all remaining parameters are fixed across runs, including the latent dimension (16), number of straight-through samples (25), $\varepsilon = 10^{-3}$, and the repulsive strength $\gamma = 0.2$. Additional details can be found in the anonymized version of source code submitted as supplementary material.

### D.5. Downstream Evaluation Details

We provide full experimental details and extended results for the two downstream stress tests of Section 4.4. Both compare three graph ensembles at $N = 256$: a generator mixture (Gen-Mix), Erdős–Rényi (ER), and NGD-generated diverse graphs (NGD). As discussed throughout the main manuscript, prior combinatorial search and generative bootstrapping methods are not computationally feasible at this node size.

**Graph ensembles.** The *Gen-Mix* ensemble draws each graph by first sampling one of 34 generator configurations uniformly at random and then sampling a graph from it. The configurations span five families: Erdős–Rényi ($p \in \{\frac{1}{16}, \frac{1}{8}, \frac{1}{4}, \frac{1}{2}, \frac{3}{4}, \frac{7}{8}, \frac{15}{16}\}$), preferential attachment with tunable attachment exponent ($m \in \{1, 2, 4\}$, additive exponent $\alpha \in \{m/2, m, 2m\}$), Holme–Kim triad formation ($m \in \{2, 4\}$, $p \in \{0.5, 1.0\}$), power-law cluster ($\gamma \in \{2, 2.5, 3, 4\}$), and stochastic block models (2–3 blocks with varied intra-/inter-block probabilities). The *ER* ensemble is $G(256, p)$ with $p \sim \mathcal{U}[0.05, 0.95]$. The *NGD* ensemble is the union of the final selected sets under the four evaluation distances (GCD, NetLSD-heat, NetLSD-wave, portrait divergence), 100 graphs each, for 400 graphs total.

**GNN regression setup.** For each target we train an independent 3-layer GIN (Xu et al., 2019b) regressor. The sole input node feature is the scalar degree. Each layer applies a 2-layer MLP (hidden width 128) followed by batch normalization and a ReLU; graph-level readout is global mean pooling fed to a 2-layer MLP head with a single scalar output. Targets are min–max normalized using statistics computed on the training set, and predictions are denormalized before computing the mean absolute error (MAE) in original units. Each model is trained on 5,000 Gen-Mix graphs with a 90/10 train/validation split, using Adam (learning rate $10^{-3}$) for 25 epochs at batch size 16 under mean-squared-error loss; the lowest-validation-loss checkpoint is retained. We evaluate on 500 held-out Gen-Mix, 500 ER, and 400 NGD graphs, and repeat over 5 trials. Graph generation and model initialization are reseeded each trial.

**Regression targets.** The six graph-level targets span spectral, connectivity, and degree structure: *spectral radius* (largest adjacency eigenvalue), *normalized number of connected components* (component count divided by $N$), *graph energy* ($\sum_i |\lambda_i|$ of the adjacency spectrum), *global efficiency* (mean inverse shortest-path distance over all vertex pairs), *degree entropy* (Shannon entropy of the degree distribution), and *average neighbor degree* (mean over vertices of the mean degree of their neighbors). Table 4 extends main-text Table 2 with per-target standard deviations across trials, and Figure 4 shows per-graph absolute-error distributions for all six targets (the main text reports three). NGD induces a heavy right tail in every target, absent under both Gen-Mix and ER.

**Greedy coloring setup.** We apply greedy graph coloring under two standard vertex orderings, largest-first and smallest-last, to the same 500 ER, 500 Gen-Mix, and 400 NGD graphs, and report both the number of colors used $k_{\text{greedy}}$ and the approximation ratio $k_{\text{greedy}}/(\Delta + 1)$ relative to the trivial Brooks bound. Table 5 reports both orderings. Results are consistent across orderings: NGD yields substantially higher mean and worst-case color counts and approximation ratios, with the worst-case NGD graphs requiring a distinct color for each vertex ($k_{\text{greedy}} = 256$), a regime not reached by ER or Gen-Mix.

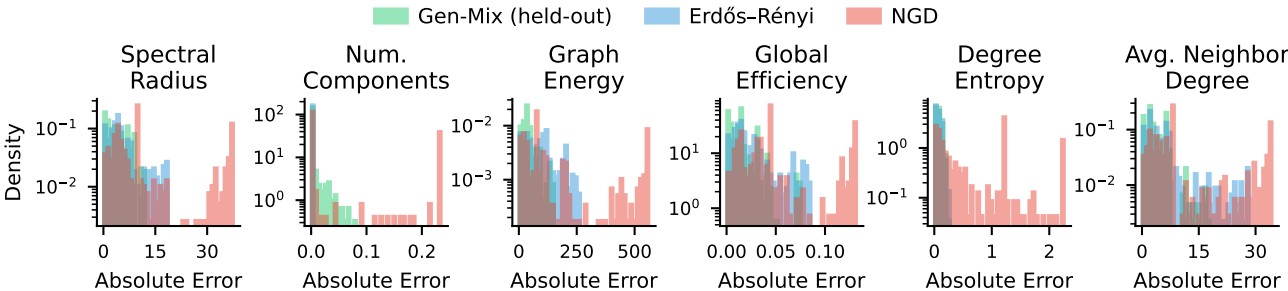

*Figure 4.* **GNN regression error distributions for all six targets.** Per-graph absolute errors for GIN regressors trained on Gen-Mix graphs ($N = 256$) and evaluated on held-out Gen-Mix, Erdős–Rényi, and NGD test sets, shown on a log-density scale (single representative trial). NGD exhibits a heavy right tail in every target, indicating structural regimes on which the learned regressors degrade sharply, absent under standard graph ensembles.

*Table 5.* **Greedy graph coloring stress test** (both vertex orderings). Greedy coloring under largest-first and smallest-last orderings is applied to 500 Erdős–Rényi graphs, 500 generator-mix graphs, and NGD-diverse graphs, all with $N = 256$. Greedy colors and approximation ratio $k_{\text{greedy}}/(\Delta + 1)$ are reported as mean $\pm$ std across each graph set; the worst (highest) value per row is bolded.

| Ordering | Metric | ER | Gen-Mix | NGD |
|---|---|---|---|---|
| Largest-first | Greedy colors | $46.24 \pm 27.89$ | $24.25 \pm 31.82$ | $\mathbf{67.87 \pm 85.39}$ |
| | Approx. ratio | $0.2991 \pm 0.0486$ | $0.2499 \pm 0.1194$ | $\mathbf{0.4796 \pm 0.2556}$ |
| | Max colors | $117$ | $129$ | $\mathbf{256}$ |
| Smallest-last | Greedy colors | $46.97 \pm 28.34$ | $24.19 \pm 32.31$ | $\mathbf{68.50 \pm 85.66}$ |
| | Approx. ratio | $0.3041 \pm 0.0494$ | $0.2367 \pm 0.1261$ | $\mathbf{0.4777 \pm 0.2563}$ |
| | Max colors | $120$ | $130$ | $\mathbf{256}$ |

## D.6. Reproducibility

All experiments use fixed random seeds. Random direction generation uses an independent fixed seed. All approaches (except IGGM) were run 5 times and the error bars using standard deviation are reported in Tables 6 and 7. IGGM was run only 3 times due to its extended runtime.

## E. Qualitative Analysis of Generated Graphs

To complement the quantitative diversity metrics, we visualize a representative slice of the graphs produced by neural graph dispersion. Figure 5 shows, for each of the four evaluation distances ($d_{\text{GCD}}$, $d_{\text{heat}}$, $d_{\text{wave}}$, $d_{\text{PD}}$), graphs drawn from the corresponding NGD-selected set at $N = 16$. Within each metric's set we exclude empty and complete graphs, remove duplicates, and then select the graph whose edge density is closest to each of five target levels ($\rho \in \{0.02, 0.05, 0.10, 0.25, 0.50\}$), so that a single row illustrates the range of structures the method spans rather than a biased sample at a single density. Node-link layouts are computed with the Fruchterman–Reingold force-directed algorithm; we caution that such layouts do not preserve the underlying graph distances, and therefore convey structural variety rather than metric geometry.

Even at this small scale, the generated graphs qualitatively exhibit substantial structural variety. They range from sparse, fragmented multi-component graphs and tree- or path-like chains to hub-and-spoke (star) topologies and dense, well-connected graphs. The graphs differ markedly in edge density, connectivity (the number and size of connected components), degree heterogeneity, and local clustering. In the shown slice, for instance, $d_{\text{heat}}$ tends to surface hub-dominated graphs while $d_{\text{PD}}$ favors elongated, low-clustering chains. This qualitative spread is consistent with the metric separation reported in the main text.

*Table 6.* Small-scale graph dispersion results with error bars under standard evaluation metrics for $N = 16$. Higher values indicate greater structural diversity. Best results per metric and graph size are shown in bold. Ablations remove either the ensemble of generators (*no ensemble*) or directional surrogate metrics (*no directions*). Parametric stochastic ensembles are run with an initial pool of $k = 100$ graphs. All other methods have the same end-to-end computation budget of 100,000 graphs. For composed methods (denoted by the $\rightarrow$ notation), the budget is split evenly among sub-component.

| Method | GCD | Heat | Wave | Portrait |
|---|---|---|---|---|
| *Parametric Stochastic Ensembles* | | | | |
| ER-mix | $4.410 \pm 0.068$ | $1.767 \pm 0.121$ | $6.001 \pm 0.136$ | $0.634 \pm 0.004$ |
| Random Graph Generators | $4.145 \pm 0.069$ | $1.805 \pm 0.060$ | $5.462 \pm 0.093$ | $0.619 \pm 0.003$ |
| *Combinatorial Search and Generative Bootstrapping* | | | | |
| ER$\rightarrow$Genetic | $7.218 \pm 0.072$ | $5.277 \pm 0.001$ | $10.629 \pm 0.098$ | $0.848 \pm 0.004$ |
| ER$\rightarrow$Genetic$\rightarrow$Local Optimization | $7.344 \pm 0.116$ | $\mathbf{5.279 \pm 0.000}$ | $10.754 \pm 0.063$ | $0.863 \pm 0.003$ |
| Greedy$\rightarrow$Genetic | $7.292 \pm 0.099$ | $5.278 \pm 0.001$ | $10.727 \pm 0.021$ | $0.853 \pm 0.007$ |
| Greedy$\rightarrow$Genetic$\rightarrow$Local Optimization | $\mathbf{7.718 \pm 0.157}$ | $5.278 \pm 0.000$ | $\mathbf{10.765 \pm 0.056}$ | $0.865 \pm 0.001$ |
| Greedy | $6.747 \pm 0.041$ | $4.686 \pm 0.033$ | $9.816 \pm 0.055$ | $0.835 \pm 0.001$ |
| Greedy$\rightarrow$Local Optimization | $7.486 \pm 0.039$ | $5.277 \pm 0.001$ | $10.671 \pm 0.049$ | $0.858 \pm 0.002$ |
| IGGM | $6.465 \pm 0.664$ | $5.227 \pm 0.000$ | $9.929 \pm 0.022$ | $0.872 \pm 0.003$ |
| *Ours* | | | | |
| Neural Graph Dispersion (NGD) | $7.621 \pm 0.108$ | $5.277 \pm 0.000$ | $10.634 \pm 0.036$ | $\mathbf{0.885 \pm 0.004}$ |
| *NGD (no ensemble)* | $6.100 \pm 0.140$ | $4.778 \pm 0.248$ | $10.140 \pm 0.021$ | $0.806 \pm 0.001$ |
| *NGD (no directions)* | $6.242 \pm 0.150$ | $5.275 \pm 0.000$ | $10.257 \pm 0.061$ | $0.823 \pm 0.002$ |

*Table 7.* Small-scale graph dispersion results with error bars under standard evaluation metrics for $N = 128$. Higher values indicate greater structural diversity. Best results per metric and graph size are shown in bold. Ablations remove either the ensemble of generators (*no ensemble*) or directional surrogate metrics (*no directions*). Parametric stochastic ensembles are run with an initial pool of $k = 100$ graphs. All other methods have the same end-to-end computation budget of 100,000 graphs. For composed methods (denoted by the $\rightarrow$ notation), the budget is split evenly among sub-component.

| Method | GCD | Heat | Wave | Portrait |
|---|---|---|---|---|
| *Parametric Stochastic Ensembles* | | | | |
| ER-mix | $5.868 \pm 0.011$ | $0.068 \pm 0.001$ | $2.646 \pm 0.018$ | $0.811 \pm 0.001$ |
| Random Graph Generators | $4.962 \pm 0.029$ | $0.905 \pm 0.040$ | $5.202 \pm 0.061$ | $0.840 \pm 0.002$ |
| *Combinatorial Search and Generative Bootstrapping* | | | | |
| ER$\rightarrow$Genetic | $7.255 \pm 0.015$ | $2.226 \pm 0.108$ | $6.154 \pm 0.151$ | $0.897 \pm 0.003$ |
| ER$\rightarrow$Genetic$\rightarrow$Local Optimization | $7.395 \pm 0.071$ | $1.235 \pm 0.156$ | $5.878 \pm 0.112$ | $0.929 \pm 0.004$ |
| Greedy$\rightarrow$Genetic | $7.760 \pm 0.010$ | $5.434 \pm 0.109$ | $\mathbf{10.974 \pm 0.105}$ | $0.959 \pm 0.001$ |
| Greedy$\rightarrow$Genetic$\rightarrow$Local Optimization | $7.887 \pm 0.007$ | $5.448 \pm 0.049$ | $10.915 \pm 0.198$ | $0.962 \pm 0.001$ |
| Greedy | $7.360 \pm 0.005$ | $2.854 \pm 0.041$ | $8.151 \pm 0.020$ | $0.941 \pm 0.000$ |
| Greedy$\rightarrow$Local Optimization | $7.781 \pm 0.009$ | $2.972 \pm 0.023$ | $8.325 \pm 0.005$ | $0.955 \pm 0.000$ |
| IGGM | — | — | — | — |
| *Ours* | | | | |
| Neural Graph Dispersion (NGD) | $\mathbf{7.981 \pm 0.091}$ | $\mathbf{5.546 \pm 0.002}$ | $10.674 \pm 0.017$ | $\mathbf{0.969 \pm 0.001}$ |
| *NGD (no ensemble)* | $7.229 \pm 0.265$ | $5.545 \pm 0.000$ | $10.448 \pm 0.049$ | $0.929 \pm 0.002$ |
| *NGD (no directions)* | $7.283 \pm 0.190$ | $\mathbf{5.546 \pm 0.001}$ | $10.565 \pm 0.044$ | $0.957 \pm 0.003$ |

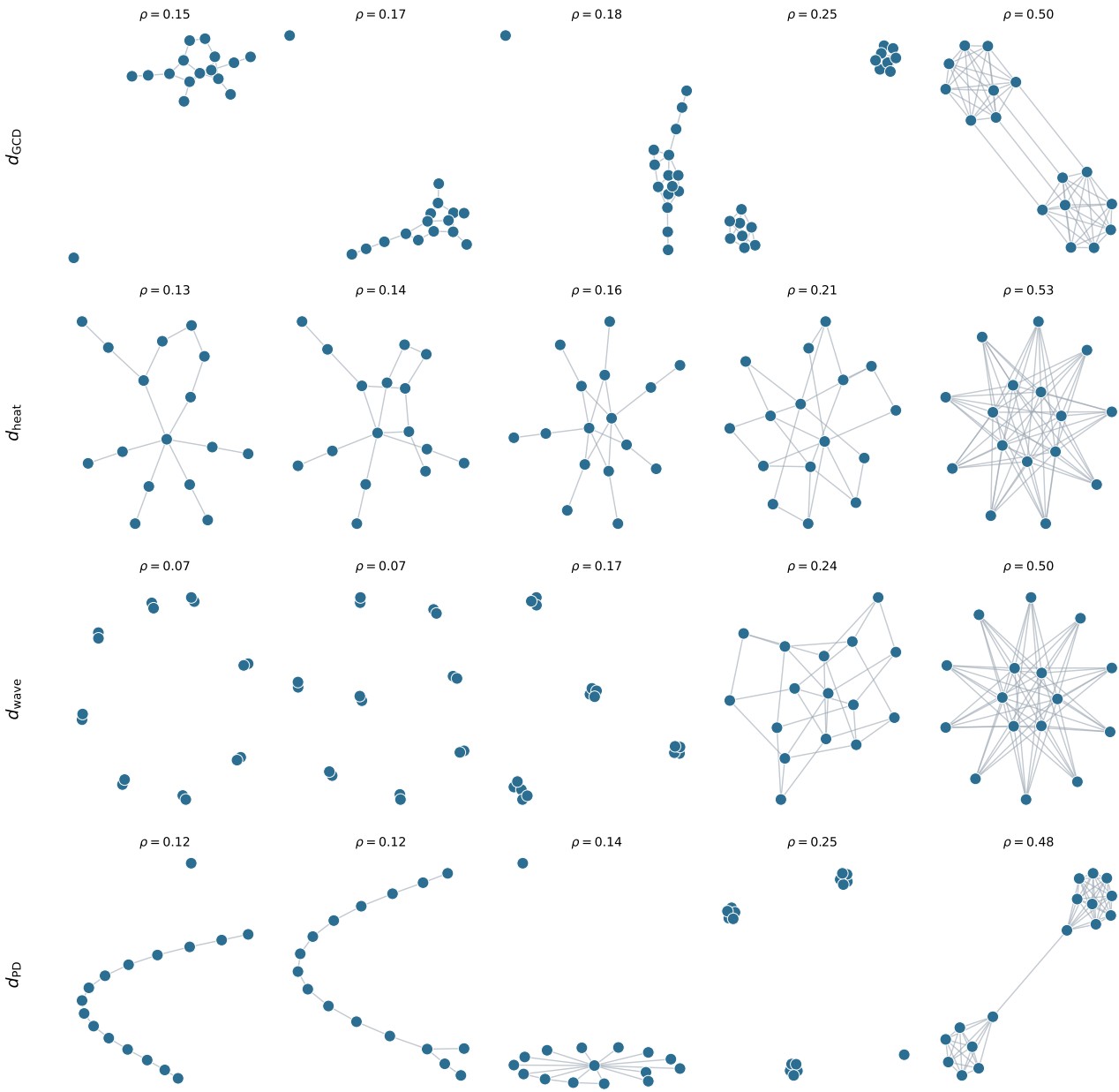

*Figure 5.* **Representative NGD-generated graphs** ($N = 16$). For each evaluation distance (rows), the graph closest to each of five target edge densities $\rho$ (columns) is shown, after excluding empty and complete graphs and removing duplicates. Node-link layouts use a force-directed algorithm and do not preserve metric distances; they illustrate structural variety, not geometry.

