# OpenReview forum: "Neural Dispersion on Graphs"
_ICML.cc/2026/Conference — ICML 2026 regular_

### Official Review · Reviewer_8jYT · 2026-03-06

**Soundness:** 3
**Presentation:** 3
**Significance:** 2
**Originality:** 3
**Overall Recommendation:** 4
**Confidence:** 4

**Summary:**

The paper proposes a method for generating a dataset of diverse graphs according to a chosen notion of diversity. The generator consists of multiple MLPs that map noise vectors to N^2 outputs, which are interpreted as edge probabilities of a relaxed adjacency matrix. Spectral features are then computed from this relaxed adjacency matrix, projected into a feature space, and the generator parameters are optimized with gradient descent to maximize pairwise distances between generated graphs.

**Compliance With Llm Reviewing Policy:**

Affirmed.

**Final Justification:**

The idea of the work is both novel and interesting.
My main concern was the motivation behind the work and its relevance to downstream applications. The authors have addressed this through additional experiments.

**Key Questions For Authors:**

- please include visualizations of generated graphs in the paper

- Why was a simple MLP generator chosen? Were GNN or transformer architectures considered?

**Limitations:**

yes

**Strengths And Weaknesses:**

## Strengths
- The paper is clearly written and easy to follow.

- the idea of directly optimizing a neural generator for output diversity is interesting

## Weaknesses
- The generator parametrization is very naive. Using an MLP that directly outputs N^2 edge scores ties the model to a fixed graph size. It would be more natural to use a GNN/transformer model that produces node embeddings and derive edge scores from them (e.g. using cosine similarity between node embeddings).

- Spectral features are computed from the relaxed adjacency matrix rather than from discrete graphs.  Methods like Gumbel Softmax could allow sampling discrete graphs while still keeping the training differentiable.

- The empirical evaluation is limited to pairwise diversity metrics between generated graphs. While these metrics quantify structural variation, they do not demonstrate that the generated graphs are useful for the applications motivating the work (e.g., benchmarking algorithms or evaluating GNNs). A downstream experiment showing that the generated sets improve such tasks compared to simple baselines would strengthen the paper.

- The paper does not include visualizations of generated graphs. The reported metrics capture only a limited set of structural properties, making it difficult to judge the qualitative behavior of the method. For example, it is unclear whether the generated graphs are typically connected or whether many samples consist of disconnected components.

---

> ### Author Rebuttal · Authors · 2026-03-31
>
> Thanks for your review and for recognizing the clarity of our presentation and the novelty of directly optimizing for output diversity. We address your concerns individually below.
>
> > The generator parametrization is very naive…
>
> We respectfully disagree that the generator architecture is unmotivated. The contribution of this work is the optimization framework for direct neural dispersion; the generator is intentionally minimal to isolate this contribution. Moreover, a GNN-based formulation is problematic in this setting: graph generation from noise requires producing structure from scratch, whereas GNNs operate on an existing input graph. Early experiments with both GNN- and transformer-based generators confirmed this difficulty empirically.
>
> Additionally, isomorphism invariance is already guaranteed by the spectral feature map (Proposition 3.3, line 286l), making equivariance constraints in the generator architecturally redundant.
>
> Regarding the fixed-size concern: dispersion on $S_N$ is by definition for fixed $N$, so this is inherent to the problem setting, not a limitation of the architecture. Despite its simplicity, the MLP generator achieves state-of-the-art dispersion across tested metrics and at scale (Tables 1–2, Figure 2).
>
> > Spectral features are computed from…
>
> We note that the training procedure already employs multi-sample straight-through estimation (Appendix D.1) [2], which, like Gumbel Softmax, is a biased gradient estimator through discrete samples. Computing spectral features on relaxed adjacency matrices during optimization is a deliberate choice for gradient stability; discrete graphs are produced at candidate selection (Section 3.4).
>
>
> > The empirical evaluation is limited to pairwise diversity…
>
> Our evaluation metrics (GCD, NetLSD, Portrait Divergence) are established in the literature as capturing meaningful structural properties [3, 4]. That said, we agree that downstream applications make the contribution more concrete. We have conducted two such experiments demonstrating the utility of NGD-diverse graphs for stress-testing both GNNs and classical combinatorial algorithms; please see our response to Reviewer 63eJ (W3) and Reviewer Fq5h (W3) for details.
>
> > The paper does not include visualizations…
>
> We address the visualization concern in our response to Reviewer Fq5h, W4; see [visualizations (stratified by density)](https://tinyurl.com/3nx3phhw) and [visualizations (random sampling)](https://tinyurl.com/cy65eer2).
>
> Regarding connectivity specifically, the proportion of connected versus disconnected graphs in the final set is a function of the evaluation metric used during candidate selection (Section 3.4), as different metrics induce different structural preferences. This is by design: the framework adapts to the user-specified notion of diversity rather than imposing fixed structural constraints.
>
> > Q1, Q2
>
> See responses above.
>
> [1] Yehudai, Gilad, et al. "From local structures to size generalization in graph neural networks." International Conference on Machine Learning. PMLR, 2021.
>
> [2] Bengio, Yoshua, Nicholas Léonard, and Aaron Courville. "Estimating or propagating gradients through stochastic neurons for conditional computation." arXiv preprint arXiv:1308.3432 (2013).
>
> [3]  Tantardini, Mattia, et al. "Comparing methods for comparing networks." Scientific reports 9.1 (2019): 17557.
>
> [4] Rozemberczki, Benedek, and Rik Sarkar. "Characteristic functions on graphs: Birds of a feather, from statistical descriptors to parametric models." Proceedings of the 29th ACM international conference on information & knowledge management. 2020.

---

> > ### Author Rebuttal · Reviewer_8jYT · 2026-04-04
> >
> > Thank you for the clarification. My main concern was the motivation behind the work and its relevance to downstream applications. The authors have addressed this through additional experiments, so I will raise my score.

---

> > > ### Author Response · Authors · 2026-04-06
> > >
> > > Thank you for the thoughtful engagement with the paper and the rebuttal. We appreciate the careful consideration of our clarifications and additional results, and the collective feedback has improved the paper considerably.
> > >
> > > We are particularly grateful for your suggestions around downstream validation and graph visualizations, which helped make the contributions more concrete.
> > >
> > > We appreciate that the originality of the core idea was recognized across all reviews, and we are grateful for the constructive criticism that pushed us to present this work as clearly and completely as possible. We believe the manuscript has been significantly strengthened through this interaction.

---

### Official Review · Reviewer_Fq5h · 2026-03-11

**Soundness:** 3
**Presentation:** 2
**Significance:** 3
**Originality:** 4
**Overall Recommendation:** 4
**Confidence:** 4

**Summary:**

The paper Neural Dispersion on Graphs addresses the problem of generating unlabelled graphs of fixed node size. The authors argue that current methods are unsatisfactory in terms of structural diversity, and propose a novel method, Neural Graph Dispersion (NGD), where generation is performed as sampling along the optimization path of a repulsive potential on the metric space of $N$-vertex graphs ($S_N$). This perspective on generating structurally dissimilar graphs is quite ingenious, as well as using optimization dynamics to act as a generator.

**Compliance With Llm Reviewing Policy:**

Affirmed.

**Final Justification:**

The authors have addressed my comments

**Key Questions For Authors:**

- Can you clarify what you mean by "strong separation" ?
- Can you discuss or even test the quality of obtained graphs ?

**Limitations:**

Even though there are evaluations and comparisons, there is no sufficient discussion on the quality of obtained graphs from a usefulness perspective.

**Strengths And Weaknesses:**

Strengths:
- The paper positions itself very clearly in the literature, and addresses the various aspects of current methods. Both strengths and limitations are discussed, such as using diffusion-based methods, random graphs etc...
- Consequently, the proposed method is genuinely novel (to my knowledge), and is principled / well-founded
- Experimentally, the authors provide exhaustive comparisons of structural diversity (with various metrics) across multiple sizes & sample sizes against current models, demonstrating the efficacy of NGDs

Weaknesses:

- The paper is quite repetitive. For example, prior work is addressed twice (Section 1 and 2) without mention of many references in Section 2. The writing is not linear, often foreshadowing a concept only to re-iterate the same thing multiple time, e.g.  the pseudometric $\phi$ is extensively (yet informally) discussed in line 136, only to be brought back in Section 3. This makes for difficult reading, when it (in my opinion) should not.
- Proposition 3.1 is slightly too informal to be such. "strong separation" is unclear, and despite the proof beign in Appendix C, i would appreciate knowing in what sense do the graphs "diverge".
- Section 5 is extremely short for any paper that introduces a new method. The only mentioned future work is extension to different combinatorial structures, but what about different dispersive potentials? Are the produced graphs actually good on certain tasks? Do they form a different class of random graphs?
- (Optional) A visualization of produced graphs along trajectories would be quite helpful, so understand better in what way they are structurally dissimilar

To summarize, I think this paper proposes a truly good concept, although the writing and presentation is unfortunate. I would not ask for more experiments (also because i wouldn't know what to ask), but I would like to see less jibberjabber in the first 5 pages, and more actual critique of the algorithm / experiments. This is not a workshop, nor is it a letter. I would expect more discussion of results, prospective work, and limitations to the method. For example, since the generator is based off of random gaussian noise, is it sensitive to Gaussian concentration in high dimensions? what about different anisotropic noise?

---

> ### Author Rebuttal · Authors · 2026-03-31
>
> Thanks for your thoughtful review and for highlighting the novelty of our generative approach, the comprehensive experiments, and the principled nature of the formulation. We address your concerns below.
>
> > W1 The paper is quite repetitive. For example…
>
> We appreciate this observation. In response to your feedback, we have performed a revision pass to reduce repetition, consolidating the prior work discussion and removing redundant descriptions of the pseudometric construction. The reclaimed space has enabled extended discussion and analysis addressed below.
>
> > W2 Proposition 3.1 is slightly too informal…
>
> In response to this feedback, we have replaced the final sentence of Proposition 3.1 with the explicit rate: "the repulsive gradient $\|\nabla_{G_i}\ell_{ij}\|$ grows as $(d(G_i,G_j)+\varepsilon)^{-(\gamma+1)}$, diverging when $\varepsilon=0$ provided $\|\nabla_{G_i} d(G_i,G_j)\|$ remains bounded away from zero." This rate was already established in the proof (Appendix C); the revision simply surfaces it in the proposition statement in order to add clarity.
>
> > W3 Section 5 is extremely short for…
>
> Section 4 already includes comprehensive baseline comparisons, scaling experiments, and ablations. We agree, however, that the discussion and future work sections can be expanded with additional analysis, and the revision pass described above has opened space to do so. We address the specific points raised below.
>
> **Downstream utility.** This is directly addressed by new experiments. We provide one example downstream experiment in our response to Reviewer 63eJ, W3, and provide another below.  We have added both experiments to the manuscript.
>
> **Alternative dispersive potentials.** Our framework is agnostic to the choice of repulsive potential; however, the application of the specific Coulomb-style objective is itself a contribution, as we show it yields stable optimization dynamics with formally characterized gradient behavior (Proposition 3.1). Early in our experiments, we evaluated alternatives including mean pairwise distance, DPP-style objectives, and $k$-nearest-neighbor repulsion, but found them to exhibit training instabilities (e.g., mode collapse or oscillatory dynamics). We have added a brief discussion of this observed behavior to the revised manuscript.
>
> **Characterizing the generated graph family.** Proposition 3.2 characterizes initialization as a latent inhomogeneous Erdős–Rényi family, but optimization trajectories escape this family by construction. Theoretically characterizing the distribution of graphs along or at convergence of these trajectories, beyond basic universal approximation arguments, is nontrivial and a genuinely interesting open question. We have added this to Section 5.
>
> **Experiment 2: Greedy Graph Coloring Stress Test.**
>
> In addition to validating GNN surrogates (Experiment 1; in response to reviewer 63eJ) diverse graph sets can also stress-test classical combinatorial algorithms. We evaluate greedy graph coloring [1] on the same three ensembles ($N=256$) as Experiment 1, measuring the approximation ratio $\rho(G) = k_{\text{greedy}}(G) / (\Delta(G) + 1)$, where $\Delta(G)+1$ is the trivial upper bound from Brook's theorem [2]. Values near 1.0 indicate worst-case performance. NGD graphs yield a mean ratio of 0.48 versus 0.24 for gen-mix ([table](https://tinyurl.com/3v48tx3n)), and the [distributional results](https://tinyurl.com/3z4da88p) reveal right tails extending to $\rho \approx 1.0$ (greedy assigns a unique color to nearly every vertex), a regime entirely absent under ER and gen-mix. These findings are consistent across vertex orderings. This demonstrates that narrow benchmarks yield misleadingly optimistic performance profiles for classical heuristics, and that NGD-diverse graphs surface worst-case behavior standard ensembles conceal. Full experimental details will be included in any camera-ready version.
>
>
> > W4 (Optional) A visualization of…
>
> We deliberately omitted graph visualizations in the original submission: standard node-link layouts depend heavily on the layout algorithm and can be misleading with respect to structural similarity in the metric space (e.g., two graphs with very different spectral properties can appear visually similar under force-directed layouts). However, based on this feedback, we have added trajectory visualizations accompanied by a disclaimer on interpretation. See [visualizations (stratified by density)](https://tinyurl.com/3nx3phhw) and [visualizations (random sampling)](https://tinyurl.com/cy65eer2).
>
>
> > Q1, Q2
>
> See responses above.
>
> [1] Matula, David W., and Leland L. Beck. "Smallest-last ordering and clustering and graph coloring algorithms." Journal of the ACM (JACM), 1983
>
> [2] Brooks, Rowland Leonard. "On colouring the nodes of a network." Mathematical Proceedings of the Cambridge Philosophical Society. Cambridge University Press, 1941.

---

> > ### Author Rebuttal · Reviewer_Fq5h · 2026-04-02
> >
> > Thank you for your answer, all my questions have been addressed (either in this rebuttal or via comments to other reviewers).
> > I will raise scores accordingly.

---

> > > ### Author Response · Authors · 2026-04-06
> > >
> > > Thank you for the thoughtful engagement with the paper and the rebuttal. We appreciate the careful consideration of our clarifications and additional results, and the collective feedback has improved the paper considerably.
> > >
> > > We are particularly grateful for your suggestions to tighten the exposition, formalize the rate in Proposition 3.1, and deepen the discussion of future directions.
> > >
> > > We appreciate that the originality of the core idea was recognized across all reviews, and we are grateful for the constructive criticism that pushed us to present this work as clearly and completely as possible. We believe the manuscript has been significantly strengthened through this interaction.

---

### Official Review · Reviewer_63eJ · 2026-03-13

**Soundness:** 3
**Presentation:** 3
**Significance:** 3
**Originality:** 2
**Overall Recommendation:** 4
**Confidence:** 3

**Summary:**

This paper studies the problem of directly generating a set of structurally diverse graphs. The authors argue that standard generative models such as VAEs or diffusion models are not well suited to this setting, because they typically require samples from a target distribution, whereas the graph dispersion problem does not naturally come with one. To address this, the paper proposes Neural Graph Dispersion (NGD), which directly optimizes diversity itself using multiple MLP-based generators trained with a Coulomb-style repulsion loss. As the distance surrogate, the method uses a differentiable, isomorphism-invariant pseudometric based on spectral moment features, and it further encourages diversity in multiple directions through random projections. During training, candidate graphs are collected from the optimization trajectory, and a final set is selected by greedy selection under the evaluation distance.

**Compliance With Llm Reviewing Policy:**

Affirmed.

**Final Justification:**

This paper provides good technical novelty and experimental studies. In the rebuttals, since they addressed my concerns, I raised my score from 3 to 4. For me, this paper is borderline, so I agree it can be accepted if there are spaces.

**Key Questions For Authors:**

See weaknesses.

**Limitations:**

There is no discussion about limitations.

**Strengths And Weaknesses:**

Strong points.

S1. The core idea is very well aligned with the problem setting.
Rather than forcing a standard distribution-learning framework onto a problem that has no natural target distribution, the paper reformulates the task so that diversity itself is optimized directly.

S2. The method is well organized and technically coherent.
The role of each component is clear: spectral-moment-based differentiable distance, directional metrics via random projections, an ensemble of generators, and bounded candidate selection at the end.

Weak points.

W1.  The method is not fully end-to-end with respect to the final evaluation metric.
During training, repulsion is optimized under a surrogate metric derived from spectral moments, while the final adjustment is done by greedy selection using the actual evaluation distance.

W2. The method is strong, but not uniformly the best.
Table 1 shows that NGD performs very well on many settings, but in some small-scale cases search-based methods such as Greedy→Genetic still achieve better results.

W3. The applicability is still somewhat limited.
The paper focuses on unlabeled, simple, undirected graphs, and the computation is based on dense-matrix operations with O(N^3) complexity.

---

> ### Author Rebuttal · Authors · 2026-03-31
>
> Thanks for your thoughtful review, for recognizing that our formulation is well suited to the dispersion problem, and for the positive assessment of our method's technical coherence. We address your concerns below:
>
> > W1
>
> We argue that this design is a deliberate feature rather than a limitation. Most evaluation metrics are non-differentiable or numerically unstable under gradient-based optimization, making a fully end-to-end approach infeasible. Moreover, optimizing under a single metric can produce degenerate configurations: e.g., maximizing GC distance favors uniform graph density, while maximizing wave kernels penalizes such extremes.
>
> Our surrogate instead induces broad structural coverage, and the selection stage aligns this to any user-specified metric with no retraining: *a single training run yields a reusable candidate pool that can be refined under competing notions of structural similarity.* We discuss this in the manuscript (lines 69-74r; 193-198r; 292-297r) and have further emphasized it in revision.
>
> > W2
>
> We note NGD is competitive with the strongest baselines at small scale and dominant at large scale, which we view as the expected tradeoff. *Notably, NGD is not trained to optimize directly for the downstream optimization metrics.*
>
> Serial exhaustive methods can achieve strong results when $|S_N|$ is small enough that a non-trivial fraction of the metric space can be explicitly explored. However, $|S_N| = 2^{\binom{N}{2}}$ grows doubly exponentially in $N$, so these methods become intractable precisely in the regime where graph dispersion is most needed.
>
> We discuss this explicitly in Section 4.1 (line 367-371r) and demonstrate it empirically in Section 4.2, where combinatorial search and generative bootstrapping methods are intractable, consistent with the scalability limitations acknowledged by their authors [1]. *NGD, by contrast, maintains high diversity across all metrics as $N$ and $k$ increase by an order of magnitude beyond what prior methods can handle.*
>
> > W3
>
> **Scope.** Our focus on simple, undirected, unlabeled graphs is deliberate: it matches the problem setting established by prior work [1], enabling direct and fair comparison. Importantly, our framework does not impose assumptions specific to this class. Extension to directed graphs is straightforward: the generator can output a full matrix $\tilde{A} \in [0,1]^{N \times N}$, and simply removing the symmetrization step yields directed edge probabilities.
>
> Regarding the concern of labeled graphs, where vertex identities reduce the symmetry group: this would in fact *simplify* the problem by eliminating the need for isomorphism-invariant representations.
>
> **Dense operations.** The $O(N^3)$ cost reflects our choice to operate over the space $S_N$ of all simple, undirected graphs on $N$ vertices. Restricting to structured subsets (e.g., sparse graphs) would reduce complexity but also constrain the metric space significantly.
>
> **Applicability.** We strongly disagree that applicability is limited. Dispersion is a well-studied problem in optimization and combinatorics [2], and graph-level dispersion has concrete applications that are referenced in the manuscript (lines 36–42l).
>
> That said, in response to this feedback, we’ve conducted two downstream application experiments that make practical utility more concrete. We describe one below, describe the other in response to reviewer Fq5h (W3), and plan to include both in any camera-ready version of the manuscript.
>
> **Experiment 1: GNN Stress Testing.**
>
> A practical application of diverse graph sets is stress-testing learned models by exposing failure modes that standard graph distributions cannot reveal, which we demonstrate here. We train independent GIN [4] regressors on 5,000 generator-mix graphs (Section 2.4, lines 164r-168l) with $N=256$ across six regression targets spanning spectral, topological, and distributional graph properties. Each model is evaluated on held-out gen-mix, Erdős–Rényi (ER), and NGD-generated test sets.
>
> Across all six targets, NGD graphs yield 3–11$\times$ higher MAE than held-out gen-mix ([table](https://tinyurl.com/75kmsk3t)). The [per-graph error distributions](https://tinyurl.com/2w2ra3vk) reveal that NGD graphs exhibit heavy right tails absent in both gen-mix and ER test sets, indicating individual graphs for which the GNN has significantly reduced predictive capacity. Full experimental details will be included in any camera-ready version.
>
> [1] Velikonivtsev, Fedor, Mikhail Mironov, and Liudmila Prokhorenkova. "Challenges of generating structurally diverse graphs." Advances in Neural Information Processing Systems 37, 2024
>
> [2] Martí, Rafael, et al. "A review on discrete diversity and dispersion maximization from an OR perspective." European Journal of Operational Research, 2022
>
> [3] Tantardini, Mattia, et al. "Comparing methods for comparing networks." Scientific reports, 2019.
>
> [4] Xu, Keyulu, et al. "How powerful are graph neural networks?." 2018

---

> > ### Author Rebuttal · Reviewer_63eJ · 2026-04-01
> >
> > Thank you very much for your rebuttals. My concerns are fully resolved, so I will raise my score.

---

> > > ### Author Response · Authors · 2026-04-06
> > >
> > > Thank you for the thoughtful engagement with the paper and the rebuttal. We appreciate the careful consideration of our clarifications and additional results, and the collective feedback has improved the paper considerably.
> > >
> > > We are particularly grateful for your suggestions around downstream utility and the discussion of scalability tradeoffs, which prompted experiments that concretely strengthened the manuscript.
> > >
> > > We appreciate that the originality of the core idea was recognized across all reviews, and we are grateful for the constructive criticism that pushed us to present this work as clearly and completely as possible. We believe the manuscript has been significantly strengthened through this interaction.

---

### Decision · Program_Chairs · 2026-04-30

**Decision:**

Accept (regular)

**Comment:**

The paper offers a graph generative model from a space of graphs with fixed set of vertices, which can maximize the diversity. Instead of learning to maximize the likelihood of underlying distributions, the work learns a generator that is optimized for certain task– which is diversity in this case. The results show that the underlying method is useful in practice.


The reviewers are modestly positive about the paper and no serious concerns were raised. The major weakness were: not being fully differentiable with respect to the final evaluation metric; do not address the common problem of generating graphs for certain benchmarking tasks; naiveness of neural parameterization, which may lead to scalability issues.